# Identification and classification of ion channels across the tree of life provide functional insights into understudied CALHM channels

**Rahil Taujale[1,2†], Sung Jin Park[3†], Nathan Gravel[1], Saber Soleymani[2], Rayna Carter[2], Kennady Boyd[2], Sarah I Keuning[3], Zheng Ruan[4], Wei Lü[3,5,6]\*, Natarajan Kannan[1,2]\***

[1]Institute of Bioinformatics, University of Georgia, Athens, United States; [2]Department of Biochemistry and Molecular Biology, University of Georgia, Athens, United States; [3]Department of Molecular Biosciences, Northwestern University, Evanston, United States; [4]Department of Biochemistry & Molecular Biology, Thomas Jefferson University, Philadelphia, United States; [5]Department of Pharmacology, Northwestern University, Chicago, United States; [6]Chemistry of Life Processes Institute, Northwestern University, Evanston, United States

**\*For correspondence:** wei.lu@northwestern.edu (WL); nkannan@uga.edu (NK)

†These authors contributed equally to this work

**Competing interest:** The authors declare that no competing interests exist.

## eLife Assessment

In this manuscript Taujale et al describe an interdisciplinary approach to mine the human channelome and further discover orthologues across diverse organisms. Further, this work provides evidence that supports a role for conserved residues in CALHM channel gating. Overall this **important** work presents findings that can be helpful to the ion channel community, as well as to those interested in improved methods for mining sequence space for their protein of interest. However, further validation of the improvements their approach shows over previous approaches is needed, making this a **solid** contribution to the literature in this field.

**Abstract** The ion channel (IC) genes encoded in the human genome play fundamental roles in cellular functions and disease, and are one of the largest classes of druggable proteins. However, limited knowledge of the diverse molecular and cellular functions carried out by ICs presents a major bottleneck in developing selective chemical probes for modulating their functions in disease states. The wealth of sequence data available on ICs from diverse organisms provides a valuable source of untapped information for illuminating the unique modes of channel regulation and functional specialization. However, the extensive diversification of IC sequences and the lack of a unified resource present a challenge in effectively using existing data for IC research. Here, we perform integrative mining of available sequence, structure, and functional data on 419 human ICs across disparate sources, including extensive literature mining by leveraging advances in LLMs to annotate and curate the full complement of the 'channelome'. We employ a well-established orthology inference approach to identify and extend the IC orthologs across diverse organisms to above 48,000. We show that the depth of conservation and taxonomic representation of IC sequences can further be translated to functional similarities by clustering them into functionally relevant groups, which can be used for downstream functional prediction on understudied members. We demonstrate this by delineating co-conserved patterns characteristic of the understudied family of the calcium homeostasis modulator (CALHM) family of ICs. Through mutational analysis of co-conserved residues altered in human diseases and electrophysiological studies, we show that these evolutionarily

constrained residues play an important role in channel gating functions. Thus, by providing new tools and resources for performing large comparative analyses on ICs, this study addresses the unique needs of the IC community and provides the groundwork for accelerating the functional characterization of dark channels for therapeutic intervention.

## Introduction

Ion channels (ICs) are membrane-bound proteins critical to many physiological processes in the human body, including regulating cell volume, neurotransmitter release, muscle contraction, and glandular secretion (*Como et al., 2021*; *Meir et al., 1999*; *Thorneloe and Nelson, 2005*; *Rajan et al., 1990*). Abnormal channel functions have been causally associated with 'channelopathies' such as Parkinson's disease, epilepsy, cardiac arrhythmia, cancer, and cystic fibrosis, to name but a few (*Edelman and Saussereau, 2012*; *Kim, 2014*; *Rivolta et al., 2020*; *Bagal et al., 2013*). The development of selective chemical probes for channels in disease states is currently hindered by the limited knowledge of the diverse gating mechanisms, ion selectivity, and pathway associations displayed by the various channels encoded in the human genome (*Bagal et al., 2013*; *Braun et al., 2020*). While IC sequences from diverse organisms encode critical information regarding underlying functions and effective mining of sequence data can provide important context for predicting and testing understudied IC functions, traditional bioinformatic approaches have had limited success due to the challenges in consistently defining the full IC complement (*Wickenden et al., 2012*; *Fodor and Aldrich, 2006*), and accurately aligning and mining large sequence datasets (*Neuwald, 2009*). Online resources like the collaborative platform Channelpedia (*Ranjan et al., 2011*), the Guide to Pharmacology (GtoP) database (*Alexander et al., 2023*), structure centric ChanFAD (*Castro et al., 2022*), and ChannelsDB (*Špačková et al., 2024*), and several other efforts have previously cataloged and curated ICs and provide extensive information on a number of IC sequences, but do not include all the human-encoded ICs, nor provide information on ICs from other organisms and taxa (*Jegla et al., 2009*; *Gao et al., 2019*; *Li and Gallin, 2004*; *Moran et al., 2015*). Most IC studies across taxa are largely confined to closely related families or limited taxonomic groups (*Lara et al., 2023*; *Uribe et al., 2024*).

The extensive diversification and widespread abundance of ICs across cell types and their similarities to other transmembrane (TM) protein families, such as transporters, have led to an inconsistent definition of the human IC complement. For example, previous literature has reported around 230 human ICs (*Jegla et al., 2009*). The Kyoto Encyclopedia of Genes and Genomes (KEGG) database has cataloged around 316 human ICs (*Kanehisa and Goto, 2000*; *Kanehisa et al., 2023*), the GtoP database lists 278 ICs (*Alexander et al., 2023*), while Pharos (*Kelleher et al., 2023*) and The Human Genome Organization (HUGO) (*Seal et al., 2023*) list around 344 and 330 proteins, respectively. Moreover, these resources classify the human ICs based on their function, which is not known for many. Furthermore, current classification rarely accounts for the pore-forming or auxiliary roles of ICs. Channel proteins often form large macromolecular complexes in association with other IC and auxiliary subunits. Separate studies have highlighted the myriad of roles that the non-conducting auxiliary subunits play in an IC complex that control channel trafficking, gating, and pharmacology (*Kato and Bredt, 2007*; *Li et al., 2006*; *Abbott, 2022*), making them suitable therapeutic targets for drugs to regulate IC activity. Unfortunately, many of the existing resources fail to identify, include, and annotate such auxiliary channels, leading to their exclusion from large-scale analyses for clinical outcomes. Due to these inconsistencies, downstream analyses have been hindered and limited to certain families or groups of closely related channels. Consequently, our current knowledge of IC functions is skewed toward a subset of well-studied 'light' channels, while a significant portion of the channelome remains understudied and is referred to as 'dark' channels by the Illuminating Druggable Genome (IDG) consortium (*Sharma and Nadler, 2021*; *Sheils et al., 2020*). Concerted efforts such as IDG and the Structural Genomics Consortium (*Abbott, 2022*) help address these gaps in knowledge by systematically identifying and prioritizing such understudied members. However, it is imperative to develop a functional and an evolutionary classification framework that provides tools for researchers to extend knowledge from light members to their dark counterparts, enabling them to make meaningful biological inferences and shed light on their critical roles in function and disease states.

The current classification of channel genes is largely based on experimentally determined electrophysiological parameters such as the type of ions they transport ($Na^+$, $K^+$, $Ca^{2+}$, $Cl^-$) and the channel

gating mechanism such as gating by voltage potential (voltage-gated) or ligand binding (ligand-gated), which has been quite helpful in placing ICs in a functional context (*Alexander et al., 2023*). One of the first comprehensive classifications of ICs was done by *Jegla et al., 2009*, where they described 235 human ICs distributed in 24 families, along with their conservation across select metazoan species, and family-specific phylogeny and evolutionary analyses. A more recent collection of ICs along with their classification can be obtained from the GtoP database (*Alexander et al., 2023*), a resource dedicated to providing detailed overviews of over 1900 human drug targets, spanning six major pharmacological targets – ICs being one of them. GtoP classifies 278 human ICs into 3 major groups (voltage-gated ion channels [VGICs], ligand-gated ion channels [LGICs], and other) and 31 families, and provides a detailed overview of these channels in a friendly interface. However, it does not include a lot of now well-known IC families such as calcium homeostasis modulator channels (CALHMs) or bestrophins (Best). Additionally, the resource being a general drug target resource is not IC-centric and lacks evolutionary and residue-level functional insights that might help hypotheses generation. The role or annotation for auxiliary channels, which play critical roles in the function and regulation of ICs, (*Kato and Bredt, 2007*; *Vacher and Trimmer, 2011*), has not been addressed in these resources either.

On the other hand, cryo-electron microscopy (cryo-EM) studies and analysis of IC structures have shown that despite extensive variation in primary sequences, several channels adopt common structural folds and mechanisms of action (*Chen et al., 2023*). For example, several ICs – potassium ($K^+$), sodium ($Na^+$), calcium ($Ca^{2+}$), transient receptor potential, and hyperpolarization-activated cyclic nucleotide-gated – grouped together as voltage-gated-like, are proposed to have evolved from a common ancestor because they share key sequence and structural features (*Yu et al., 2005*; *Huffer et al., 2020*). Thus, quantitative comparisons of these commonly conserved regions at the sequence and structural level can provide new residue-level insights into IC function and a stronger basis for functional classification, much like in other large protein families such as kinases and glycosyltransferases (*Shrestha et al., 2020*; *Picado et al., 2020*; *Taujale et al., 2020*; *Taujale et al., 2021*). This is imperative since selective targeting of channels in diseases and a mechanistic understanding of disease-associated mutations in channelopathies will require a residue-level understanding of function-determining features in both well-studied 'light' and understudied 'dark' channels. Thus, there is a critical need for an updated classification that places ICs in functionally and evolutionarily related groups, enabling a deeper residue-level analysis of light and dark channels alike.

To develop a comprehensive classification of ICs, we first mined literature and sequence data sources to collect information on ICs stored across various databases and in disparate data sources and formats to build a well-curated knowledgebase toward defining the human ion 'channelome.' A comprehensive list encompassing established and putative IC sequences in the human proteome is provided as a unified table alongside other contextual data for functional annotations such as regulatory interactions, ion selectivity, and pore-lining residues. Leveraging the identification of a pore region and pore-lining residues, we make a clear distinction between pore-containing vs auxiliary ICs. Based on this curation, we then perform a comprehensive classification of ICs into well-defined groups and families and identify the extent of understudied channels in each family, providing a premise for prioritizing studies in IC families with more understudied members.

We further performed a comprehensive evolutionary analysis to define and collect more than 48,000 well-defined IC orthologous relationships from diverse taxonomic groups, spanning metazoans, fungi, protists, bacteria, and archaea. We demonstrate the application of these sequences in annotating understudied CALHMs at residue-level resolution using Bayesian statistical approaches (*Neuwald, 2014*) and in predicting and experimentally validating the structural and functional impact of disease-associated mutations. These studies support our working premise that integrative mining of available sequence, structure, and functional data on ICs from diverse organisms (well-studied and understudied) will provide an important context for defining sequence and structural features associated with understudied channel functions.

## Results

## Defining the IC complement of the human genome using informatics approaches

We first sought to collect and curate the full IC complement across the human proteome by systematically mining all the existing data sources, collating information from literature, and running various informatics tools (see Methods) to identify related sequences based on overall similarity and pore-defining regions. In brief, we first collected all the sequences listed as ICs in the comprehensive UniProt database (*UniProt Consortium, 2018*), as well as Pharos (*Kelleher et al., 2023*), KEGG (*Kanehisa and Goto, 2000*; *Kanehisa et al., 2023*), GtoP (*Alexander et al., 2023*), and HUGO Gene Nomenclature Committee (HGNC) (*Seal et al., 2023*) databases and subjected them to a series of annotations as described in *Table 1*. The complete annotation table for all human ICs is available in *Supplementary file 1A*. Primarily, we mined the literature for functional and structural annotations on these sequences. Broadly, we have included six annotation categories. (1) The identifier labels include the UniProt ID, name, and Target Development Level (TDL) designation from Pharos (as of July 3, 2025). The IDG consortium, through Pharos, assigns TDL, which can be Tclin, Tchem, Tbio, or Tdark, based on the literature data available for proteins in terms of their potential as drug targets. Including this designation in the annotation is especially relevant for identifying and prioritizing understudied ICs for further investigation. (2) The classification labels describe the Group, Class, and Family the IC falls into. This information has been mined from literature including UniProt, Pharos, GtoP, and HGNC, and the field 'Family designation' provides a consensus family label for the IC. (3) The functional labels have been manually mined from various sources, relying mostly on previous literature in the 'Lit Resource' column. An important column in this section is the 'Unit' column, which describes whether the IC is directly involved in forming the pore region, where it can be pore-forming (directly involved in forming an ion-conducting pore), two-pore (contains two tandem pore-conducting regions), or auxiliary (does not have its own pore domain but interacts with other pore-conducting IC subunits as part of a complex) (please see below and Methods for further definition of an auxiliary IC). We also provide the ions, gating mechanism, and UniProt functional description in this annotation category. To verify the ion and gating mechanism annotations, we mined the literature using large language models (LLMs) and retrieval augmented generation (RAG) systems (Methods, *Figure 1—figure supplement 1*). The RAG system was able to successfully verify about 40% of the ion and gating mechanism annotations, while for others, it was unable to find supporting evidence, either because no evidence was available in the existing literature or no relevant references were found. The annotations that were not verified by the RAG system are indicated with an asterisk in *Supplementary file 1A*. (4) The structure-related category includes the PDB ID of any experimentally generated crystal structure or an Alpha-Fold ID for a computationally predicted structure of the IC. (5) The Complex/Interaction label provides information on the types of complex the IC is involved in forming, along with the interactors. (6) The last section for TM and pore domain-related labels has been curated extensively using various sources and predictors. The TM regions and pore-containing functional domains are annotated based on literature references and prediction tools, as outlined in the Methods section. TM predictions by TMHMM (*Krogh et al., 2001*) and Phobius (*Käll et al., 2007*) are provided alongside the TM annotations provided in UniProt. First, we supplemented this information by adding TM information based on the literature review and the source. Then, we further analyzed any ICs with an experimentally determined structure using the MOLE software (*Sehnal et al., 2013*) to identify and annotate the pore region and pore-lining residues. The MOLE software starts by defining the membrane region of the protein, followed by the identification of cavities and computation of the pore boundaries to predict the pore-lining residues. These predictions are used as additional evidence for defining the pore- containing functional domain region and a pore-containing IC sequence. Sequences without an identified pore region were classified as auxiliary, as defined by *Gurnett and Campbell, 1996*, to provide functional context for this classification. A snapshot of the list of annotations we collected through this process is shown in *Table 1*, using Aquaporin-1 as an example.

In the final phase, we conducted sequence similarity searches to detect sequences exhibiting significant homology with any curated sequences, subsequently subjecting these candidates to all annotation procedures to determine their eligibility as ICs. By combining the results from this annotation process, we were able to achieve the following: (1) compile a list of human ICs based on the presence of distinct TM regions, a detectable pore, and evidence of ion conductivity, (2) distinguish

**Table 1.** List of features annotated for the collected ion channel (IC) sequences.

The labels are grouped by their annotation category. Labels for Aquaporin-1 are shown as examples of each annotation label.

**Identifier labels**

| | |
|---|---|
| UniProt | P29972 |
| Name | Aquaporin-1 |
| Symbol | AQP1 (CHIP28) |
| Target Development Level (Pharos) | Tbio |
| Length | 269 |

**Classification labels**

| | |
|---|---|
| Family designation | Aquaporin |
| Group | Other |
| Class | |
| Family | Aquaporin |
| Subfamily | |

**Functional labels**

| | |
|---|---|
| Unit | Pore-containing |
| Ion | Water |
| Gate mechanism | Ligand-gated (cGMP) |
| Lit Resource | PMID:26365508, PMID:16962972 |
| UniProt function | Form water-specific channel/plasma membranes of red cells and kidney proximal tubules |

**Structure-related labels**

| | |
|---|---|
| PDB ID | 8CT2 |
| AlphaFold ID | |

**Complex/Interaction-related labels**

| | |
|---|---|
| Auxiliary | No |
| Characterized domains | MIP |
| Auxiliary domain | no |
| Auxiliary protein | |
| Notable interactors | EPHB2 |

**TM and pore domain-related labels**

| | |
|---|---|
| Pore domain start | 8 |
| Pore domain end | 228 |
| Domain length | 220 |
| Does it pass through membrane at least once | Yes |
| # of TM domains (predicted by TMHMM) | 6 |
| # of TM domain (predicted by Phobius) | 6 |
| TM organization | 1\|2\|3\|4\|5\|6 |
| MOLE pore residue first | 35 |
| MOLE pore residue last | 185 |
| # of TMs | 6 |
| # of Transmembranes (TMs)+Intramembranes (IMs) | 10 |

*Table 1 continued on next page*

*Table 1 continued*

**TM and pore domain-related labels**

| | |
|---|---|
| TMstart (UniProt) | 8 |
| TMend (UniProt) | 228 |
| TMsList | T:8–36,T:49–66,I:71–76,I:77–84,T:95–115,T:137–155,T:167–183,I:187–192,I:193–200,T:208–228 |
| TM Lit Resource | PMID:10644652 |
| Lit based TMstart_1 | 8 |
| Lit based TMend _1 | 36 |
| Lit based TMstart_2 | 49 |
| Lit based TMend_2 | 66 |
| Lit based TMstart_3 | 95 |
| Lit based TMend_3 | 115 |
| Lit based TMstart_4 | 137 |
| Lit based TMend_4 | 155 |
| Lit based TMstart_5 | 167 |
| Lit based TMend_5 | 183 |
| Lit based TMstart_6 | 208 |
| Lit based TMend_6 | 228 |

between a pore-containing IC vs an auxiliary IC, (3) identify novel putative IC sequences, and (4) curate and classify the full IC complement in the human proteome. These results are summarized in *Figure 1* and *Supplementary file 1A*.

419 human IC sequences were curated using this approach, representing an increase of 75 sequences compared to the previous consensus of 344 ICs in humans (*Bagal et al., 2013*; *Kelleher et al., 2023*). A comparison of the curated IC sequences in this study against the list of ICs in other resources – KEGG, GtoP, and Pharos, is provided in *Figure 1—figure supplement 2* and *Supplementary file 1B*. These sequences were classified into 4 major groups – VGICs, LGICs, chloride channels, and others – defined previously (*Neuwald, 2009*) and 55 families. VGICs constitute the largest group, comprising 186 sequences distributed across 21 families, followed by LGICs with 82 sequences in 10 families. Of the 419 sequences, 62 could not be assigned to the four major groups and were categorized into outlier families. Among these, 28 are pore-containing ICs, with 19 of the 28 distributed across four families (CALHM, otopetrin [OTOP], TM channel-like (TMC), and tweety homolog [TTYH]). These families, collectively referred to as 'Unclassified' families, are labeled as such in *Figure 1*. The remaining nine pore-containing sequences could not be assigned to a distinct family and were grouped together under a single 'Unclassified' family. Based on the evaluation of the pore-containing regions, 343 out of 419 ICs were annotated as pore-containing ICs, while the remaining 76 were classified as auxiliary and fell into 17 different families. 23 of these auxiliary ICs are soluble with no detectable TM domains. Any auxiliary ICs that did not belong to a distinct family were collectively classified into an 'Auxiliary unclassified' family.

Next, we sought to use the curated set to define similarities across IC families. Traditional sequence-based bioinformatics approaches, such as pairwise sequence alignment (e.g. BLASTp) (*Camacho et al., 2009*) or profile hidden Markov models (e.g. HMMER) (*Eddy, 1998*), are limited in their ability to detect relationships among human ICs due to the extensive divergence in their primary sequences. These methods often fail to identify homologous ICs across different families, as the sequence similarity falls below detectable thresholds. Consequently, such approaches are inadequate for comprehensive classification or comparison of IC families in humans. Instead, we relied on representations (embeddings) derived from evolutionary scale protein language models (*Lin et al., 2023*) to capture sequence, structure, and evolutionary information, and use them to generate pairwise sequence alignment. Specifically, for the 343 pore-containing ICs, we passed their pore-containing

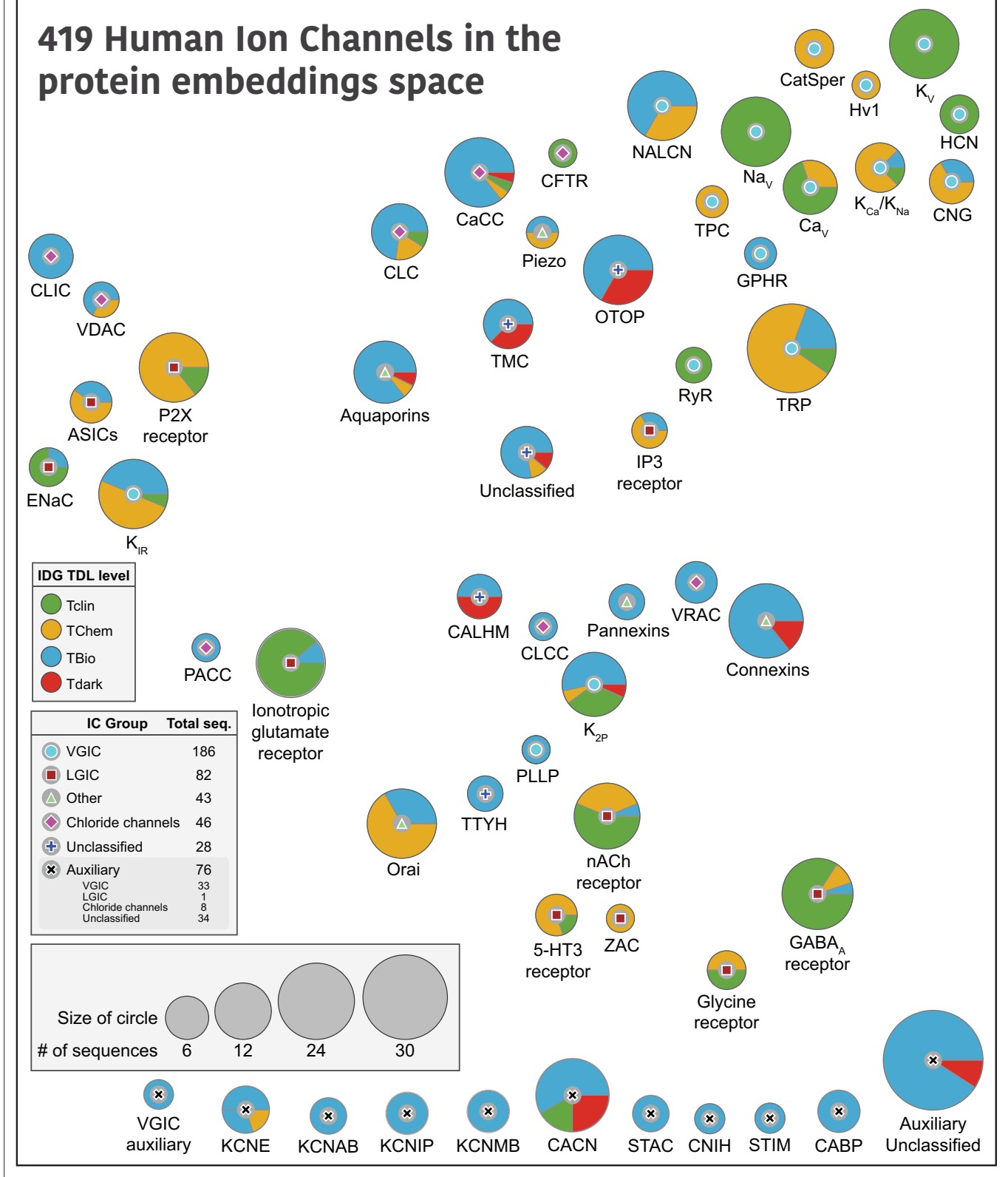

**Figure 1.** Distribution of human ion channels (ICs) across different families. Each circle represents a human IC family, with the symbol at the center indicating its Group. The size of the circles is proportional to the number of sequences in that family, and the colored pies indicate the proportion of their Target Development Level (TDL) status as designated by Illuminating Druggable Genome (IDG). The placement of the bubbles is based on the average coordinates of all the members within that family in a distribution of uniform manifold approximation and projection (UMAP) embeddings

*Figure 1 continued on next page*

*Figure 1 continued*

generated using protein embedding-based pairwise sequence alignments. An embedding-based sequence alignment approach was used to overcome the vast divergence of IC sequences with minimal sequence similarity. A family-level abstraction was done to provide an intuitive view undeterred by the relationships of individual ICs across families. A detailed view of the full UMAP plot showing the placement of individual ICs is provided in *Figure 1—figure supplement 3* for reference. Auxiliary IC families at the bottom were not part of the embedding-based analysis due to the lack of a pore-containing domain, and thus placed arbitrarily at the bottom of the figure.

The online version of this article includes the following figure supplement(s) for figure 1:

**Figure supplement 1.** Retrieval augmented generation (RAG) annotation pipeline used for validating the annotation of ion specificity and gating mechanism.

**Figure supplement 2.** Upset plot showing the overlap of ion channel (IC) sequences based on their UniProt IDs across the current study, the Kyoto Encyclopedia of Genes and Genomes (KEGG) database, Guide to Pharmacology (GtoP), and Pharos.

**Figure supplement 3.** Uniform manifold approximation and projection (UMAP) embeddings plot of all human ion channel (IC) sequences.

---

functional domains to a protein sequence embedding model called DEDAL (*Llinares-López et al., 2023*) to perform a pairwise sequence alignment. Given that sequence similarity is largely restricted to the pore-containing functional domains of ICs, we computed protein embeddings using only these regions. As auxiliary ICs lack such domains, they were excluded from this part of the analysis. The resulting all-vs-all sequence similarity scores were used to generate uniform manifold approximation and projection (UMAP) embedding scores (*Figure 1—figure supplement 3*). An average of all IC sequences within a family was calculated to place the IC family bubbles in *Figure 1*. Thus, families placed close to each other in *Figure 1* indicate their similarities in sequence embeddings, which capture learned representations of sequence, structure, evolutionary conservation, and functional properties. Since the auxiliary IC families did not have UMAP embedding scores, they are arbitrarily placed at the bottom of the figure.

Most of the VGIC families group together on the top right except for inward rectifier potassium channels ($K_{IR}$), two-pore domain potassium channels ($K_{2P}$), and plasmolipin (PLLP), where $K_{2P}$ and PLLP are more centrally dispersed, while $K_{IR}$ falls closer to LGIC families, acid-sensing ion channels, epithelial sodium channel (ENAC), and P2X purinoceptors, all of which share similar TM topology with only two TM helices. The inositol 1,4,5-trisphosphate and ionotropic glutamate receptors are also centrally located, while the Cys loop receptor LGIC families group separately from others toward the bottom right, indicating their shared functional and structural similarities. Interestingly, pannexins, volume-regulated anion channels (VRAC), connexins, chloride channel CLIC-like (CLCC), and CALHM families group closer together centrally, indicating shared features.

We also mapped TDL annotations defined by the IDG (*Sharma and Nadler, 2021*) in Pharos (*Kelleher et al., 2023*) to highlight the depth of understudied ICs within each group, and hopefully, spark additional studies of those members. Briefly, IDG describes Tclin as targets that have an approved drug; Tchem has known high potency small molecule binding targets; Tbio has experimentally supported Gene Ontology (GO) (*Ashburner et al., 2000*) annotations based on published literature, while Tdark is manually curated at the primary sequence level in UniProt, but does not meet any of the Tclin, Tchem, or Tbio criteria (*Kelleher et al., 2023*). There are 19 Tdark, 185 Tbio, 88 Tchem, and 127 Tclin ICs spread across all families. The proportion of these levels for each family is shown as pie charts in *Figure 1*. The CALHM family has the highest proportion of Tdark ICs, with OTOP, TMC, connexins, calcium-activated chloride channels (CaCC), calcium voltage-gated channel (CACN), $K_{2P}$, and the unclassified families accounting for the remaining Tdark ICs, emphasizing the need for additional research on these families. On the other hand, VGICs – voltage-gated potassium channel ($K_V$), voltage-gated sodium channel ($Na_V$), voltage-gated calcium channel ($Ca_V$), ryanodine receptor (RyR) – and LGICs, including gamma-aminobutyric acid type A ($GABA_A$) receptor, nicotinic acetylcholine (nACh) receptor, and ENAC, have the highest proportion of Tclin ICs, solidifying their status as the well-studied families.

## Mining and cataloging IC orthologs across the tree of life

Next, we sought to extend this annotation and collect related ICs from other organisms across different taxonomic lineages from the tree of life (*UniProt Consortium, 2018*) using the full complement of the annotated human IC sequences. To this end, we used a graph-based orthology inference approach (*Huang et al., 2021*) starting from the full-length and the pore-containing functional domains of the

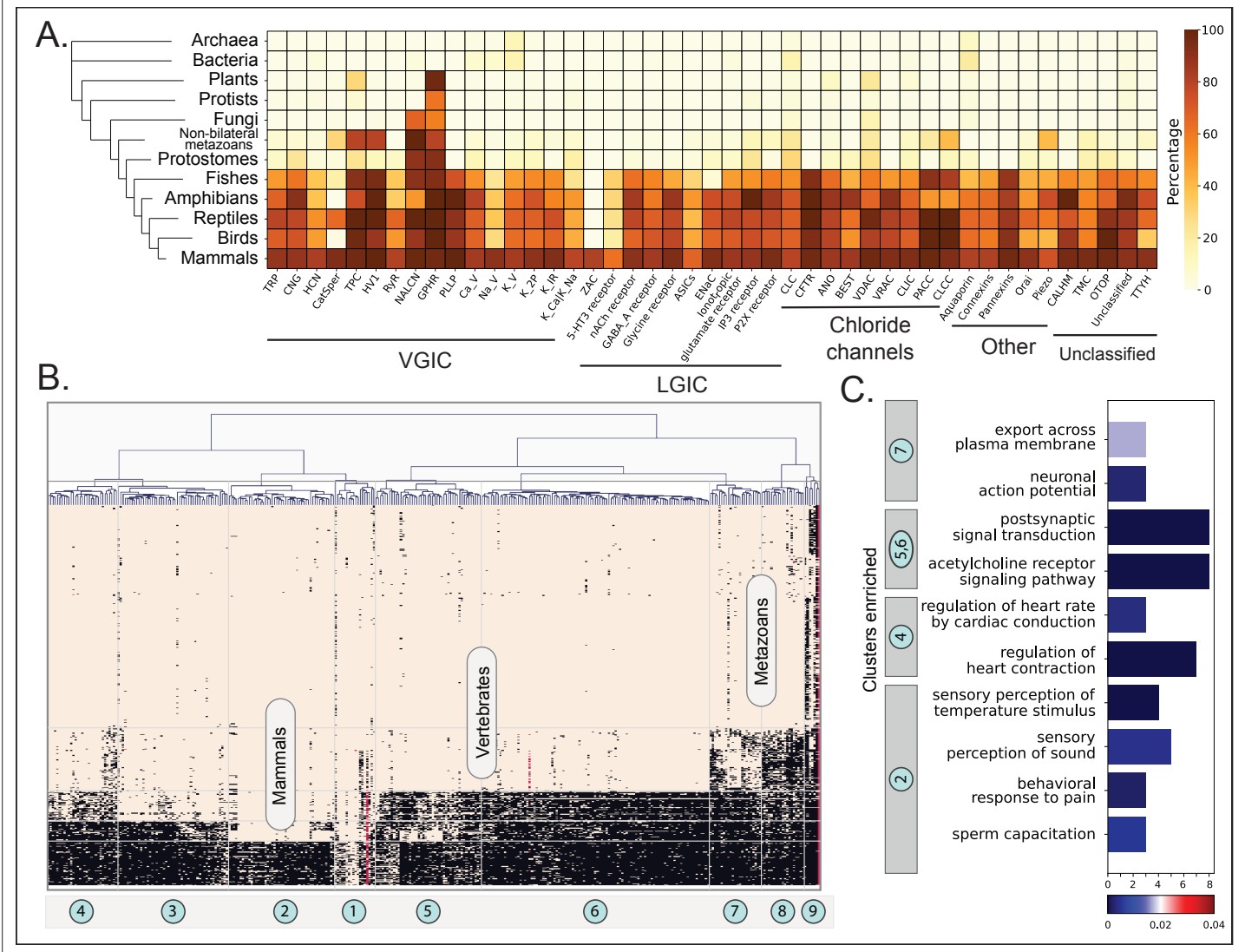

**Figure 2.** Orthology profiling of human ion channels (ICs). (**A**) Heatmap showing the percent of orthologs detected for each IC family within a given taxonomic lineage. The taxonomic groups are shown in the vertical axis with a tree on the left. Darker color represents a higher percentage of orthologs detected. Percentages were calculated as (total number of orthologs found for all ICs in a family)/(total number of organisms queried in the taxonomic lineage * number of sequences in the family). (**B**) Clustergram depicting the presence/absence of orthologous sequences of ICs across eukaryotic taxonomic lineages. ICs are clustered along the horizontal axis into nine distinct clusters. Taxonomic groups are shown on the vertical axis. Each square in the heatmap is colored based on the orthology relationship found for a specific IC in a specific organism (black: one-to-one ortholog present, red: co-ortholog detected, brown: no orthology detected). (**C**) Results from the enrichment analysis performed on human ICs of each cluster. The x-axis shows the number of ICs in the cluster enriched for the Gene Ontology (GO) term shown on the y-axis. The bars are colored based on their FDR values for the enriched term. For a full list of enriched terms, please refer to **Supplementary file 1D**.

human IC sequences to define orthologous relationships across more than 1500 proteomes from the UniProt Proteomes database (**UniProt Consortium, 2018**). These selected proteome datasets include 353 Archaea, 696 Bacteria, and 547 Eukaryota organisms. This method has previously been successfully employed to define orthology across such a large collection of proteomes for protein kinases (**Huang et al., 2021**), and since it relies on both the full-length sequence and the annotated pore-containing domains, it can more accurately define true orthologs that are indeed evolutionarily related across organisms. Since this method relies on the pore-containing domain annotations for a domain-based orthology inference, we only used the 343 pore-containing IC sequences for this analysis, and did not include the auxiliary ICs that lack a pore-containing domain.

By combining domain and full-length searches, we could identify more than 48,000 IC orthologous relationships across diverse organisms. *Figure 2A* shows a heatmap depicting the phylogenetic profiling of human IC orthologs along the horizontal axis across the taxonomic lineages shown by the tree along the vertical axis. The color intensity indicates the percentage of orthologs detected for all the sequences in the IC family for a given taxonomic group. The highest number of orthologs were found for Aquaporins, including bacterial and archaeal orthologs, indicating that this family of ICs is the most conserved across lineages. Along with Aquaporins, the two paralogs of Golgi pH regulator ICs (GPHRs), both indicated as Tbio channels, were found to have the largest number of orthologs across metazoans. Along with GPHRs, several other ICs, notably members of the $K_V$ family and $GABA_A$ receptors, display some of the most widespread orthologs across organisms. Some dark channels, such as members of the TMC (TMC7 and TMC3) and CALHM (CALHM5 and CALHM3) families, are well conserved across metazoans and vertebrates, respectively, yet very little is known about their functions. On the other hand, some Tdark channels, such as potassium channel K member 7 (KCNK7) and TMC4, have a lineage-specific set of orthologs that extend only up to mammals. A full list of all the orthologs detected through this analysis is provided in *Supplementary file 1C*.

To use this evolutionary conservation for functional inference of dark ICs, we first used hierarchical clustering to group ICs with similar orthology profiles into related clusters, which led to the definition of nine clusters, as shown in *Figure 2B*. Since the orthology profiles in prokaryotic lineages were very sparse, only the eukaryotic orthology profiles were retained for this analysis. Within eukaryotes, each cluster has a well-defined signature of orthology conservation. For example, ICs in cluster 2 have most of their orthologs only in mammals, whereas clusters 4, 5, and 6 have orthologs spanning other vertebrates. Similarly, clusters 7 and 8 have orthologs extending to other metazoans, while seven ICs in cluster 9, including the two Golgi pH regulators A and B and the sodium leak channel NALCN, have detectable orthologs in fungi, plants, protists, and other eukaryotic lineages. To translate these patterns of orthology profiles to function, we performed a functional GO term enrichment analysis for the human ICs in each cluster (*Figure 2C*). As expected, the most significant GO terms were related to IC function (*Supplementary file 1D*). However, beyond channel function, the enriched functional GO terms obtained for each cluster correlate well with the physiological functions present in the orthologous organismal groups. For example, clusters 4, 5, and 6 are conserved in vertebrates and have six dark IC sequences, including unclassified ICs such as CALHM3 and 5, and connexins GJA10, GJD4, and GJE1, which had functional enrichment for regulation of heart contraction and heart rate, traits that could be closely related to a closed blood circulatory system with an endothelium present in vertebrates (*Monahan-Earley et al., 2013*). On the other hand, cluster 2 with orthologs in mammals was conserved in mammalian-specific reproductive functions such as sperm capacitation.

## Using the orthology information to identify evolutionary constraints in CALHMs

Once the orthologs have been identified across different organisms, they can be leveraged to find evolutionarily conserved signals that could point to functional similarities within related groups of sequences. To this end, we classified subsets of orthologous IC sequences into evolutionarily related clusters using a Bayesian partitioning with pattern selection (BPPS) algorithm, which classifies sequences based on patterns of amino acid conservation and variation in a large multiple sequence alignment (see Methods) (*Neuwald, 2014*). For this analysis, we focused on the orthologs of the CALHM family of IC sequences. The CALHM proteins constitute a family of large pore channels, forming oligomeric assemblies of different sizes (*Ma et al., 2016*). It has six member sequences in humans (CALHM1–6), three labeled as dark channels, constituting one of the families with the highest prevalence of dark ICs. It has been well established that CALHM1 is activated by removing extracellular $Ca^{2+}$ and membrane depolarization and that the heteromeric CALHM1 and CALHM3 channels are implicated in the ATP release during taste perception (*Ma et al., 2018*). CALHM1 has also been shown to regulate cortical neuron excitability (*Ma et al., 2012*), locomotion, and induces neurodegeneration in *Caenorhabditis elegans* (*Tanis et al., 2013*). On the other hand, CALHM6 has been shown to be important for immune system functions by facilitating induction of immune cells during infection (*Danielli et al., 2023*). In contrast, the activation stimuli and physiological roles of other family members remain largely unexplored. All CALHM family members share a common arrangement in the TM domain, with each subunit consisting of four TM segments (S1-S4), forming a large cylindrical

pore lined by the first TM segment S1 along with a short N-terminal helix preceding S1. The exact gating mechanism of the CALHM channels is still unclear, partly since all their cryo-EM structures have been in an open state. However, previous studies have indicated that the N-terminal region, called the amino-terminal helix (NTH), plays a crucial role in modulating voltage dependence and stabilizing the closed channel state (*Tanis et al., 2017*), while a more recent study has suggested that the voltage-dependent gate is formed by the proximal regions of S1 (*Ma et al., 2025*). Collectively, these studies indicate the role of the amino termini – the NTH and S1 regions, to play a crucial role in the gating mechanism and the determination of pore size (*Choi et al., 2019*) of CALHMs.

Thus, we performed a phylogenetic analysis to find evolutionarily conserved residues that might shed more light on these critical mechanisms that govern CALHM function. *Figure 3A* shows a phylogenetic tree depicting the evolutionary relationship across the six members of the CALHM family across diverse taxa. CALHM1 and 3 form a distinct clade from CALHM2, 4, 5, and 6. We first analyzed 5805 CALHM homologs to identify pattern positions conserved across all these sequences with the hypothesis that such conserved positions could point to shared functional features across all CALHMs. The Bayesian analysis identified 13 aligned positions conserved across all 6 CALHM homologs, which we will refer to as CALHM shared patterns (*Figure 3B and C*). Most of these conserved positions were hydrophobic amino acid residues, and five conserved cystine residues, four of which are involved in forming inter-molecular disulfide bridges (C46=C130, C48=C162). Residue position numbers reflect the numbering based on human CALHM2 (PDB id: 6uiv). Interestingly, five of the conserved positions (F44, Y56, I61, P64, W117) are located close to the amino termini in a functionally important linker region connecting S1 and S2 (S1-S2 linker). Specifically, this linker region could regulate the dynamic conformational changes of S1, where the S1 could adopt either a vertical conformation relative to the membrane plane, resulting in an enlarged pore size, or a lifted conformation, leading to a reduced pore size (*Figure 3D*; *Choi et al., 2019*). Therefore, mutations in this linker are expected to affect channel functions. Based on the positioning of these conserved residues, and previous studies that highlight the importance of this region in gating functions, we hypothesized that these conserved pattern positions play a role in the gating mechanism of CALHM. The conservation of these residues across orthologs of all six CALHM sequences further suggests that all CALHM paralogs could share this gating mechanism.

To test our hypothesis and determine the functional importance of the CALHM shared patterns, we sought to perform a series of mutational experiments to determine the functional implications of perturbations at these positions. To achieve this, we methodically determined target mutations for each position. We first scanned the Genome Aggregation Database (gnomAD) (*Gudmundsson et al., 2022*) to check for any prevalent variations at these conserved positions within the sampled population to use as our mutational targets. We found several disease variants at these positions that are listed in *Figure 3E* that were used to prioritize target mutations. For positions where a variation was not found, we tested their significance by performing alanine mutations, causing a deletion of the side chain at the β-carbon.

## Targeted mutational and electrophysiological studies of CALHM conserved residues

To assess the functional roles of the predicted conserved residues in CALHM channels, we performed targeted mutational and electrophysiological analyses in two representative human CALHMs: the well-studied human CALHM1 and the relatively understudied human CALHM6.

Building on the insights from previous structural and functional studies (*Ma et al., 2025*; *Choi et al., 2019*), which implicated the NTH/S1 region in channel gating, we prioritized the constraints located in the S1-S2 linker and at the interface between S1 and the TM domain (*Figure 3B, C, and E*). These two regions were hypothesized to play distinct roles in gating: the S1-S2 linker as a flexible hinge and the S1-TMD interface as a stabilizing contact for S1 movement.

Both total protein western blotting and surface biotinylation assays showed that the mutants have either comparable or substantially higher expression levels than wild type, confirming that neither protein expression nor trafficking to the plasma membrane was impaired (*Figure 4—figure supplement 1*). We then established electrophysiological measurements for wild-type CALHM1 and CALHM6 following protocols described in the Methods. Our results were consistent with previous findings (*Ma et al., 2012*; *Figure 4*). In addition to room temperature, which is commonly used for patch-clamp

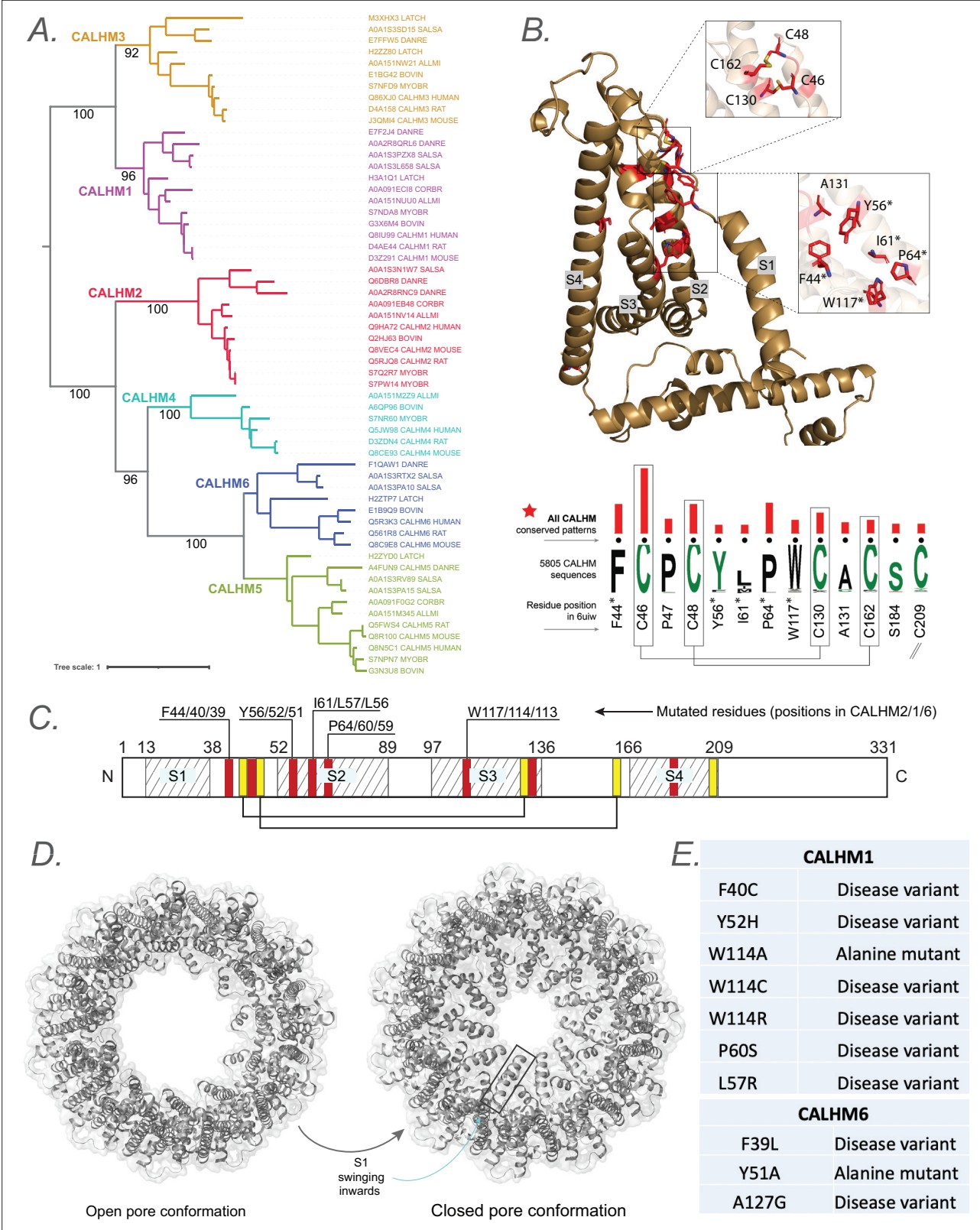

**Figure 3.** Evolutionary analysis of calcium homeostasis modulator (CALHM) reveals conserved pattern positions. (**A**) A phylogenetic tree depicting the evolutionary relationships across all six CALHM members with orthologs across different taxa. (**B**) The conserved pattern positions conserved across all six CALHM members are shown as a weblogo with the red bars indicating the significance (longer bars indicate higher significance) of conservation and are mapped into a representative structure of human CALHM2. The four transmembrane helices are labeled S1-S4. Conserved residues that

*Figure 3 continued on next page*

*Figure 3 continued*

are targeted for mutation are highlighted using an asterisk symbol in the labels. (**C**) Schematic representing the location of transmembrane regions and identified conserved pattern positions in a representative human CALHM2 sequence. The residues targeted for mutations are labeled with their corresponding positions for CALHM2, 1, and 6 shown in the labels. (**D**) Cartoon representation of CALHM2 structure (PDB ID: 6uiv and 6uiw) in open and closed conformation, respectively. (**E**) List of disease variants and mutations performed in the conserved pattern positions for functional studies.

The online version of this article includes the following figure supplement(s) for figure 3:

**Figure supplement 1.** Conserved pattern positions identified within the clade for calcium homeostasis modulator (CALHM)2, 4, 5, and 6.

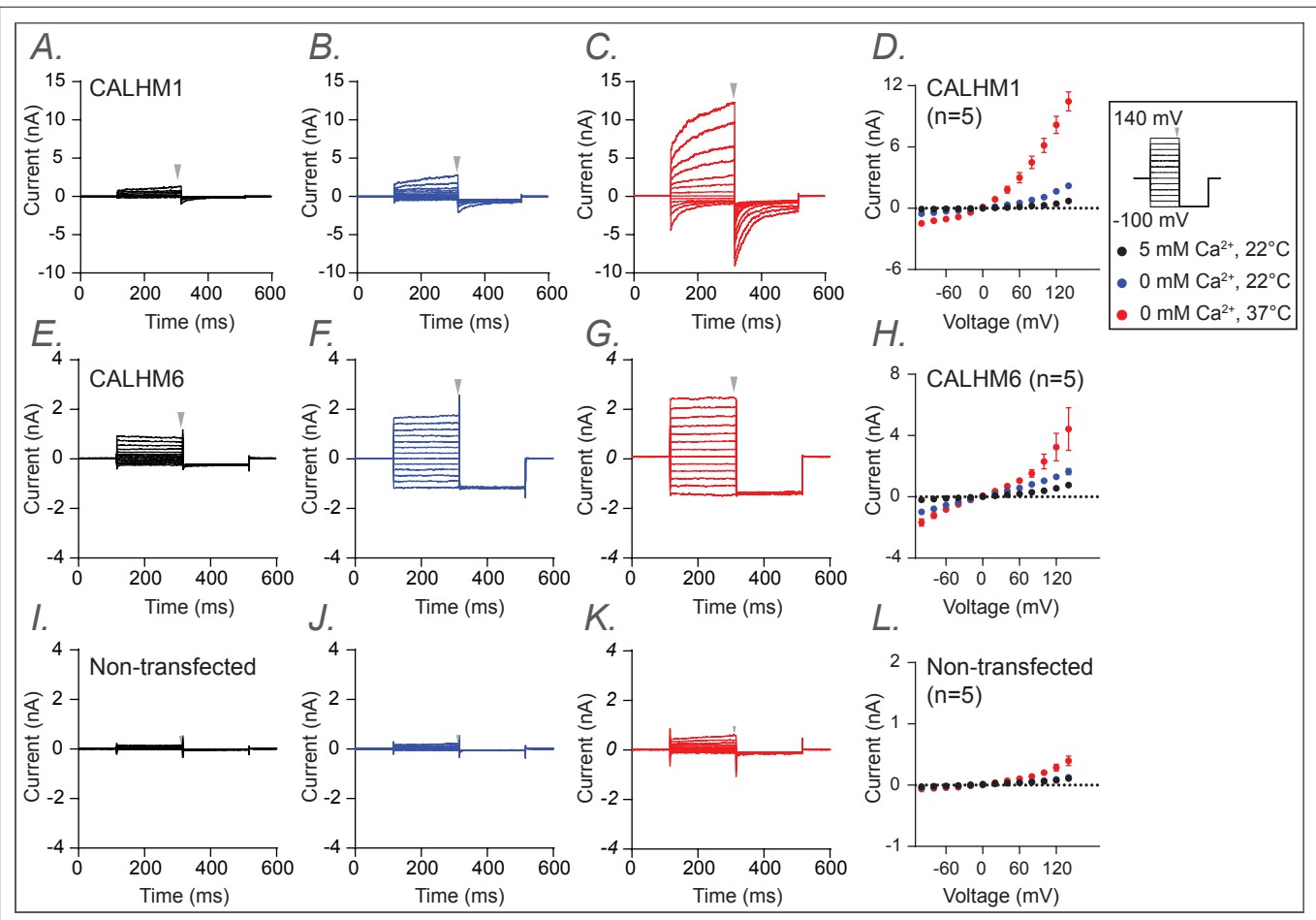

**Figure 4.** Electrophysiological studies of human calcium homeostasis modulator CALHM1 and CALHM6. Whole-cell voltage-clamp recordings were performed in tsA cells overexpressing wild-type CALHM1 (A–D) and wild-type CALHM6 (E–H), as well as in non-transfected cells (I–L). Currents were measured under three conditions sequentially from the same cell: 5 mM $Ca^{2+}$ at 22°C (A, E, I; black), 0 mM $Ca^{2+}$ at 22°C (B, F, J; blue), and 0 mM $Ca^{2+}$ at 37°C (C, G, K; red). Voltage steps ranged from −100 mV to +140 mV, followed by a final tail pulse at −100 mV, with a holding potential of 0 mV (protocol illustrated in the box on the right). Current-voltage (I–V) relationships were plotted in D, H, and L using mean current amplitudes (averaged across independent cells) measured at the end of the voltage steps. The arrow indicates the time point at which current amplitudes were measured. Error bars represent SEM (D, n=5; H, n=5; L, n=5).

The online version of this article includes the following source data and figure supplement(s) for figure 4:

**Figure supplement 1.** Expression analysis of wild-type calcium homeostasis modulator CALHM1, wild-type CALHM6, and their mutants.

**Figure supplement 1—source data 1.** PDF file containing original western blots for *Figure 4—figure supplement 1*, indicating the relevant bands and treatments.

**Figure supplement 1—source data 2.** Original files for western blot analysis displayed in *Figure 4—figure supplement 1*.

studies of CALHM channels, we also conducted measurements at physiological temperatures (37°C). We observed that CALHM1 activity was significantly higher at physiological temperature than at room temperature, as reported previously (*Kwon et al., 2021*; *Jeon et al., 2021*), while the CALHM6 currents showed only a small increase (*Figure 4*). Notably, both CALHM1 and CALHM6 currents were inhibited by extracellular $Gd^{3+}$ (*Figure 5C and I*), a commonly used inhibitor for the CALHM family, consistent with previous studies (*Ma et al., 2012*; *Danielli et al., 2023*; *Ma et al., 2025*). The robust currents at 37°C, particularly those of CALHM1, provide a solid basis to interpret the phenotypes of the mutants tested, as detailed below.

Mutations of a predicted conserved residue in the S1–2 linker (F40 in CALHM1; F39 in CALHM6) either abolished or markedly reduced channel activity in both CALHM1 and CALHM6, presumably by impeding the conformational dynamics of S1 required for channel gating (*Figure 5P and Q*; *Figure 5—figure supplement 1*). Similarly, mutation of a conserved tyrosine residue on S2 (Y52 and Y51 in CALHM1 and CALHM6, respectively), whose side chain directly contacts the S1-S2 linker, also resulted in strong phenotypic changes: Y52A in CALHM1 abolished channel activation (*Figure 5P and Q*; *Figure 5—figure supplement 1*), while Y51A in CALHM6 converted the channel from voltage-independent to voltage-dependent gating, resulting in outward rectification (*Figure 5J–L, P, and Q*; *Figure 5—figure supplement 1*). These residues are likely key determinants controlling the conformational dynamics during gating.

Interestingly, mutations in other conserved residues, near – but not within – the S1–2 linker (W114, L57, and P60 in CALHM1; W113 and A127 in CALHM6), also abolished or markedly reduced channel activity (*Figure 5P and Q*; *Figure 5—figure supplement 1*). Among these, W114 in CALHM1 and W113 in CALHM6 directly contact S1. Thus, substituting these bulky hydrophobic residues with smaller (cysteine or alanine) or positively charged (arginine) residues is expected to alter the conformational dynamics of S1 and therefore impair channel gating. Notably, however, the CALHM6 W113A was an exception, retaining wild-type-like currents at 37°C and exhibiting voltage dependence and inhibition by extracellular $Gd^{3+}$ similar to wild-type CALHM6 (*Figure 5M–O*), suggesting that this mutation does not impair fundamental gating properties.

Finally, since most mutants showed either reduced or completely abolished activity, we included a positive control in the electrophysiological experiments: I109W on CALHM1, which has been previously documented to increase channel activity. As expected, we reproduced this gain-of-function phenotype (*Figure 5*). To further explore channel-specific differences, we also examined the corresponding mutant in CALHM6 (L108W). Interestingly, L108W decreased CALHM6 channel activity (*Figure 5*), possibly reflecting inherent differences between the two channels.

## Discussion

Here, for the first time, we have computationally defined the full IC complement of the human genome – the 'channelome'. We provide this comprehensive list and a rich annotation of functional, structural, and sequence features, mapping them to 4 widely accepted IC groups and further classifying them into 55 families. We also highlight unclassified outlier groups that contain most of the understudied 'dark' and potentially novel unclassified IC sequences. As part of our curation, we also provide annotation for the pore-containing domain that is based on multiple sources, including an extensive literature review, further strengthened by the application of literature mining using LLMs and structural analysis tools. We use the pore-containing domain to distinguish pore-containing from auxiliary ICs, providing additional functional context. Our annotations can serve as a reference for comparing ICs within or across families, designing experiments for functional studies, and identifying potential targets to illuminate the functional relevance of understudied ICs further.

The application of LLMs, specifically RAG systems, in our research demonstrates significant potential in automating literature mining and correcting certain inaccuracies. However, the system currently faces limitations related to diverse data presentation formats, complex channel subunit relationships, and variant gene nomenclature, which hinder its effectiveness. The high frequency of responses where evidence was not found underscores the need for enhanced entity recognition capabilities, particularly for accurately identifying gene and protein names across different nomenclatures. Additionally, the RAG system's ability to critically engage with data by identifying and amending presumed inaccuracies can be widely used to verify and correct annotation entries, though this also raises concerns about potential overcorrections or misinterpretations without adequate validation. By more data cleaning

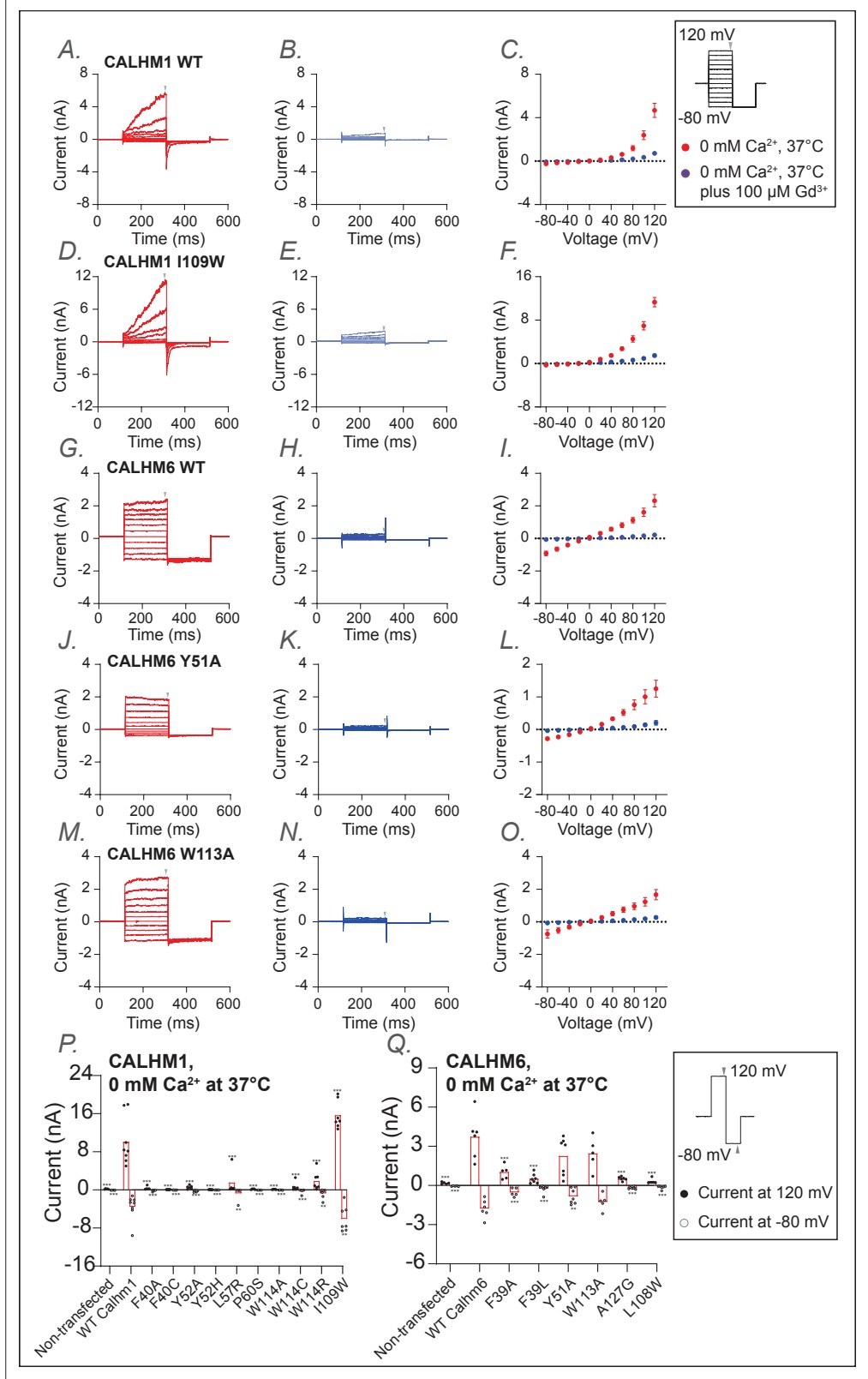

**Figure 5.** Functional characterization of calcium homeostasis modulator CALHM1 and CALHM6 mutants at conserved residues at 37°C. Whole-cell voltage-clamp recordings were performed in tsA cells overexpressing wild-type CALHM1 (**A–C**), CALHM1 mutant I109W (**D–F**), wild-type CALHM6 (**G–I**), and CALHM6 mutants (Y51A (**J–L**) and W113A (**M–O**)). Currents were measured under two conditions: 0 mM Ca²⁺ at 37°C (A, D, G, J, M; red)

*Figure 5 continued on next page*

*Figure 5 continued*

and 0 mM $Ca^{2+}$ at 37°C plus 100 µM $Gd^{3+}$ (B, E, H, K, N; blue), with both conditions recorded sequentially from the same cell. Voltage steps ranged from −80 mV to +120 mV, followed by a final tail pulse at −80 mV, with a holding potential of 0 mV (protocol shown in the box on the right). Current-voltage (I–V) relationships were plotted in C, F, I, L, and O using mean current amplitudes (averaged across independent cells) measured at the end of the voltage steps. The arrow indicates the time point at which the current was measured. The number of independent measurements (cells) were: n=5 (**C**); n=5 (**F**); n=5 (**I**); n=5 (**L**); n=5 (**O**). Error bars represent SEM. (**P, Q**) Current amplitudes obtained using a two-step voltage protocol (from +120 mV to −80 mV; protocol shown in the box on the right) are compared between wild-type CALHM1 and its mutants (**P**), and between wild-type CALHM6 and its mutants (**Q**). Each dot represents an independent measurement (cell), and bar represents the mean current amplitude across cells. The number of independent measurements (cells) for each bar in P and Q are shown from left to right: 5, 8, 5, 8, 6, 7, 5, 5, 5, 6, 7, 7 (**P**); 5, 6, 5, 7, 7, 5, 6, 6 (**Q**). Statistical analysis was performed using one-way ANOVA with Bonferroni's post hoc test, comparing each mutant to wild type (*p<0.05; **p<0.01; ***p<0.001).

The online version of this article includes the following figure supplement(s) for figure 5:

**Figure supplement 1.** Functional characterization of calcium homeostasis modulator (CALHM)1 and CALHM6 mutants at conserved residues at 22°C.

and standardizations, LLMs and RAG systems can become more reliable and integral tools for curating and expanding bioinformatics databases, ultimately enhancing the accuracy and comprehensiveness of our IC knowledgebase.

To the best of our knowledge, identifying true orthologs across the tree of life performed here is the first large-scale effort to consolidate IC orthologs across diverse organisms and provides a comprehensive view of the evolutionary conservation of individual ICs. Based on this analysis, we could pinpoint the Aquaporins family of ICs as the most ancient family of ICs with orthologous sequences present back to bacterial and archaeal species. More importantly, the clustering of human ICs based on their depth of conservation placed them into functionally meaningful clusters with many enriched functions exclusive to their orthologous taxonomic group. The placement of understudied ICs in such groups allows for meaningful functional predictions, which can be further elucidated using experimental approaches.

The orthologous sequences identified and presented here can serve as a valuable resource for performing evolutionary and functional analysis using statistical and machine learning approaches. We demonstrate one such use by performing a Bayesian pattern-based classification on the CALHM subset of ICs to identify amino acid positions significantly conserved across all CALHM sequences. These features reside on a functionally important S1-S2 linker region that potentially governs their gating mechanism. Because these features were conserved and shared across all CALHM sequences, we also hypothesize that all CALHM sequences share this gating mechanism. By performing targeted mutations on these conserved residues, we show that a mutation in any of these evolutionarily conserved residues results in a dramatic loss of gating function, thus highlighting the functional importance of the identified pattern positions.

Among the conserved positions targeted for mutations, mutations of residues in the S1-S2 linker (F40C in CALHM1; F39L in CALHM6; Y52A in CALHM1; and Y51A in CALHM6) resulted in strong phenotypic changes, most likely due to their direct involvement in the conformational dynamics of S1 for channel gating. Mutations in W114 in CALHM1, which is not directly on the S1-S2 linker, also resulted in abolished activity; however, mutations in an analogous position W113 in CALHM6 did not impair channel function, presenting an exception compared to other residues across CALHM1 and 6. Among other residues that are near – but not within – the S1-S2 linker, the positioning of P60 on the S2 helix of CALHM1 is particularly intriguing. In the cryo-EM structure, P60 resides approximately midway along S2, inducing a distortion or bending of the helix due to the rigid ring structure of proline. This bent configuration allows extensive contact between the extracellular portion of S2 and the S1-S2 linker, while the intracellular portion of S2 interacts extensively with S1. Furthermore, we hypothesize that this bent conformation of S2 contributes to the flexibility of the S1-S2 linker, as if S2 were straightened, it would consequently straighten the S1-S2 linker, potentially reducing its flexibility. Replacing the proline with glycine eliminates the structural constraint imposed by proline's cyclic side chain, allowing S2 to adopt a more regular, uninterrupted helical structure. Consequently, a straightened S2 may lead to reduced flexibility of the S1-S2 linker, ultimately impacting channel

gating. Similarly, L57 in CALHM1 is positioned near P60 on the protrusion of the bend of S2, with its side chain facing residues on the adjacent S3. We hypothesize that the larger side chain of the L57R mutant would force S2 to straighten due to steric collision between the bulky arginine side chain and residues on S2, again reducing the flexibility of the S1-S2 linker and altering channel gating. A127 in CALHM6 resides in a short alpha helix within the extracellular domain. While A127 does not directly interact with the S1-S2 linker, it is adjacent to the highly conserved disulfide bond (between C41 and C126), linking the helix containing A127 to the S1-S2 linker. Thus, it is conceivable that the A127G mutant may indirectly affect the conformation of the S1-S2 linker through this disulfide bond.

These findings complement prior studies and provide additional insights into CALHM gating mechanisms. Several previous studies examined conserved residues in CALHM channels and proposed roles in gating (*Tanis et al., 2013*; *Danielli et al., 2023*; *Tanis et al., 2017*; *Ma et al., 2025*; *Kwon et al., 2021*; *Syrjänen et al., 2023*). These studies were based on comparisons of a limited set of homologs from model organisms such as *C. elegans* or mouse and identified residues in various parts of the protein that influence gating, providing important clues toward their functioning. Here, we have extended the analysis to encompass thousands of CALHM sequences collected from the entire tree of life, allowing us to identify residues conserved across all family members through evolution. Guided by a hypothesis from previous structural and functional studies (*Ma et al., 2025*; *Choi et al., 2019*), which highlighted the NTH/S1 region as a key element in channel gating, we focused on evolutionarily conserved residues in the S1-S2 linker and at the interface of S1 with the rest of the TMD. We reasoned that if S1 movement is critical for gating, then these two structural elements – the S1-S2 linker, acting as a hinge, and the S1 interface with the TMD, serving as a stabilizing contact – would be key determinants of the conformational dynamics of S1.

Together, our experimental data indicate that the conserved residues at the S1-S2 linker and its immediate surroundings play an important role in CALHM channel gating, supporting a model in which S1 dynamics are central to the gating mechanism (*Ma et al., 2025*; *Choi et al., 2019*). Furthermore, the contrasting phenotypes of Y52A in CALHM1 vs Y51A in CALHM6, W114A in CALHM1 vs W113A of CALHM6, and I109W in CALHM1 vs L108W in CALHM6 highlight differences in gating properties specific to each family member. Overall, the complementary results from this study and the published literature highlight the complexity of CALHM gating and suggest that distinct conserved elements contribute to both shared and lineage-specific gating mechanisms in this unique family of large-pore channels. Importantly, this was demonstrated not only for the well-studied CALHM1 but also for the relatively understudied human CALHM6, providing valuable clues into shared gating features across light and dark CALHM sequence sets.

In addition to the CALHM shared patterns, we also present a second subset of conserved positions shared only within CALHM2, 4, 5, and 6, paralogs that are evolutionarily closer compared to CALHM1 and 3 (*Figure 3—figure supplement 1*). These subsets of conserved residues fall in the intracellular helical segment that is involved in oligomerization and formation of the pore complex governing CALHM function. Previous studies have indicated that CALHM2 adopts a unique undecameric oligomer different from the CALHM1 octamer (*Syrjanen et al., 2020*). Thus, with the presence of shared patterns within the subset of CALHM2 and the understudied CALHM4, 5, and 6 in this intracellular helical region, we hypothesize that these conserved positions help maintain a similar mode of oligomerization between these evolutionarily related subsets of CALHMs. Because many disease mutations map to these conserved positions, we believe the datasets and approaches offer a powerful avenue for elucidating the functional and clinical relevance of the understudied ICs.

## Methods
### Identification and annotation of human ICs
The annotation of human ICs was performed using a semi-automated pipeline. First, all the protein sequences that were labeled as ICs were collected from the UniProt (*UniProt Consortium, 2018*), KEGG (*Kanehisa and Goto, 2000*; *Kanehisa et al., 2023*), Pharos (*Kelleher et al., 2023*), GtoP (*Alexander et al., 2023*), and HGNC (*Seal et al., 2023*) databases. The annotations described in *Table 1* were then collected from literature sources manually and compiled together. The UniProt-specific labels and the complex formation information were extracted from the UniProt database. The sequences were run through TMHMM (*Krogh et al., 2001*) and Phobius (*Käll et al., 2007*) to

predict TM and helical regions. The predictions were cross-referenced and confirmed against the CDD (*Lu et al., 2020*), Pfam (*Mistry et al., 2021*), PrositePattern, PrositeProfiles (*Henschel et al., 2007*), and Simple Modular Architecture Research Tool (SMART) (*Letunic et al., 2021*) databases, where a reference was available. Finally, the prediction of the pore region and pore-lining residues was supplemented using the MOLE software (*Sehnal et al., 2013*). The prediction of the pore region and the information available in previous literature was combined to annotate the final set of auxiliary IC sequences. An auxiliary IC is defined as any sequence that itself does not have a pore domain, but has experimental evidence of being part of an IC complex. Thus, any IC where a pore domain could not be found was subjected to additional literature review to find evidence for their interactions with the pore-containing ICs, and if the evidence was found, they were included as auxiliary ICs; otherwise, they were removed from our curated IC list.

## RAG system for verifying ion selectivity and gating mechanism annotations

To systematically evaluate the ion selectivity and gating mechanism fields in the curated human IC dataset, we built a RAG pipeline comprising three sequential stages: corpus construction and vectorization, query formulation and similarity search, and evidence synthesis with an LLM. All scripts and configuration files are archived in the project repository (https://github.com/esbgkannan/ionchannels-final-pdf, copy archived at *Kannan, 2026*).

### Corpus construction and vectorization

PubMed identifiers (PMIDs) linked to each IC were obtained from our annotation pipeline. Full-text PDFs were downloaded, converted to plain text with the *pdfminer.six* Python library, and segmented into overlapping fragments of approximately 1000 tokens (50-token overlap) using the *Character-TextSplitter* module of LangChain. Each fragment was embedded with the OpenAI *text-embedding-3-large* model (3072-dimensional vectors). Embeddings and fragment metadata (PMID, page number, fragment index) were stored in a local *Qdrant* vector database. Prior to embedding, boilerplate headers, footers, and reference lists were removed to retain only article body text.

### Query formulation and similarity search

For every IC, two natural-language questions were generated automatically: (1) 'Is there any evidence that <*ION*> is the ion selectivity of the <*IC-NAME*>>ion channel?', and (2) 'Does this article provide evidence for <*GATING*> as the gating mechanism for the <*IC-NAME*>>ion channel?'.

Here, <*IC-NAME*>>encompasses the UniProt primary name and all recognized synonyms (produced by extract_alternative_names.py script in https://github.com/esbgkannan/ionchannels-final-pdf). Each query was submitted to Qdrant to retrieve relevant fragments from the database. When PMIDs were available in the dataset, an initial PMID-filtered search was performed; filtered and unfiltered results were subsequently merged and fragments with the highest cosine similarity retained.

### Evidence synthesis with an LLM

The retrieved fragments were inserted into a fixed prompt and supplied to GPT-4o (temperature = 0). The model was instructed to return a JSON object having three keys and values of answer (Found, or Not Found), confidence (a number between 0 and 1), and evidence (an explanation of LLM's decision for the answer). Outputs were parsed with LangChain's *JsonOutputParser*, combined with fragment metadata, and compiled to a single JSON file for further analysis. Predictions with confidence ≥0.80 were accepted automatically; lower-confidence or *Not Found* results triggered manual review. The structured records were merged with the annotation table, enabling automatic comparison with pre-existing entries and flagging of discrepancies for curator review.

Entries for which no supporting evidence could be located (model answer 'Not Found' confirmed on manual inspection) are marked with an asterisk in *Supplementary file 1A*. An illustrative exchange is provided in *Figure 1—figure supplement 1*.

### Defining a pore containing functional domain

We aimed to define the pore-containing functional domain so that it spanned the pore region and included all the TM domains present in an IC. This ensured that our domain definition included the

pore region and any other functionally important TMs, while excluding any accessory domains on the flanking sequences. To ensure we did not miss any TM regions, four different sources of annotation information were combined to define the pore-containing functional domain for the 343 ICs. (1) The TM predictions from TMHMM (*Krogh et al., 2001*) and Phobius (*Käll et al., 2007*) were first used to identify the TM regions. (2) The UniProt-based TM annotations were matched with the predictions to verify the TM regions further. (3) The literature-based TM annotations were then used to verify the TM positions and organization, where available manually. (4) Where experimentally resolved crystal structure coordinates were available, the MOLE software was used to identify the pore-lining residues.

## Pairwise sequence alignment using sequence embeddings

The pore-containing functional domains for the 343 pore-containing ICs were passed to DEDAL (*Llinares-López et al., 2023*), which was run using standard parameters and resulted in homology logit scores based on an all-vs-all pairwise sequence alignment using their sequence embeddings. These scores were used to generate a sequence similarity matrix passed to the umap function of the umap_learn package v0.5.7 (*McInnes et al., 2018*) in Python 3.9 to generate 2D UMAP embeddings. This was used to generate a 2D scatterplot that defined the placement of individual ICs shown in *Figure 1—figure supplement 3*. Finally, an average position of ICs within a family was used to define the placement of that family in *Figure 1*.

## Orthology detection and analysis

The KinOrtho pipeline (*Huang et al., 2021*) was used for defining the orthologs and co-orthologs of human ICs across the tree of life. KinOrtho employs a graph-based orthology inference approach using both the full-length and the pore domain regions for inferring orthologous relationships, primarily relying on sequence similarity within these regions. The pipeline began with pairwise sequence similarity searches using both full-length and pore-containing domain sequences of human ICs against the target proteomes database. Since a pore domain needs to be defined for this analysis, we only used the 343 pore-containing IC sequences to perform orthology detection. They were used as queries for running KinOrtho against a reference database of curated canonical proteomes from the UniProt Proteomes Release 2022_05 that consisted of 343 Archaea, 696 Bacteria, and 548 Eukaryota proteomes. The initial sequence similarity search was conducted using BLASTp (*Camacho et al., 2009*) with default parameters and an e-value cutoff of 1e-5 to retain high-confidence hits. The top hit from this forward search was then used as a query in a reciprocal BLASTp search against the human proteome, applying a more stringent e-value threshold of 1e-200 to retain only the top and high-confidence hits. Pairs of sequences were retained only if each was the top hit for the other in this reciprocal search. Such a reciprocal best-hit strategy minimizes false positives by ensuring that the candidate orthologs are each other's most similar sequence in the respective proteomes, which is particularly important when distinguishing true orthologs from paralogs or other homologs that may share partial similarity but have diverged functionally (*Hernández-Salmerón and Moreno-Hagelsieb, 2020*).

All the retained hit pairs were then passed to cluster analysis using OrthoMCL (*Li et al., 2003*), followed by filtering of relationships to keep only relationships within the same cluster. This process was conducted separately using both the full-length human IC sequences and their pore domain sequences. Only sequences that were identified as orthologs in both the full-length and domain-based analyses were retained as high-confidence orthologs. This ensured that both overall sequence similarity and conservation of functionally critical domains were satisfied.

Finally, the obtained ortholog sequence sets were subjected to a series of validation tests that provide more guardrails and ensure we avoid duplicates, fragments, and extraneous hits, ensuring that only sequences with high-quality annotation were retained as true orthologous relationships. These validation tests followed the following steps:

Step 1: Check UniProt Entry Type: We first check the 'entryType' flag in UniProt to check whether it is: 'Reviewed' or 'Unreviewed'. 'Reviewed' sequences are manually annotated and reviewed, thus are of the highest quality and safe to include. Any sequences with this status are passed. Sequences with 'Unreviewed' status proceed to Step 2.

Step 2: Check for Protein Existence Evidence: For the 'Unreviewed' sequences, check their 'proteinExistence' flag. This flag can have one of five values: (1) Evidence at protein level. (2) Evidence

at transcript level. (3) Inferred from homology. (4) Predicted. (5) Uncertain. Flags 1, 2, and 3 are dependable evidence of protein existence, thus are passed, whereas 4 and 5 are low confidence and subjected to further verification in Step 3.

Step 3: For the proteins with low confidence (4 or 5) protein existence values, we next check five different measures/flags to ensure its integrity:

1. Check their sequence version for unusual patterns. Multiple updates with high sequence versions or a very old version update indicate unstable or deprecated sequences, respectively.
2. Perform sequence level checks.
    a. Check for compositional bias: Sequences with compositional bias might be low complexity regions or repeats.
    b. Check for proportion of non-standard amino acids: Presence of a large number of non-standard amino acids could also indicate poor protein annotation.
    c. Check sequence length: Unusually short (<30 aa) or long (>5000 aa) sequences could indicate fragments or fusion errors and misannotation.
3. Check for cross-references: If the protein completely lacks annotated domains, or additional cross-referenced metadata, it might suggest poorly characterized or erroneous proteins. For any sequence that is subjected to Step 3, it should pass all the checks in this step to be included as an orthologous hit.

48,694 unique orthologous relationships from 36,846 sequences passed the orthology pipeline and validation checks and were included in the final list of orthologous sequences. The list of all the orthologous sequences that passed the validation check is provided in *Supplementary file 1C*.

To illustrate the extent of these orthologous sequences across different taxonomic lineages, they were used to create a presence/absence matrix of orthologs, with rows representing each human IC and columns representing the different organisms. To reduce individual granularity and get estimates at the family and lineage level for visualization, first, the orthologs were grouped by IC families and then, by their defined taxonomic lineages. For each group representing one IC family and one taxonomic lineage, a percentage value representing the proportion of detected orthologs was calculated using the following: (total number of orthologs found for all ICs in a family)/(total number of organisms queried in the taxonomic lineage * number of human IC sequences in the family). These percentages are depicted in the heatmap in *Figure 2A*, where each cell represents an IC family and its proportion of orthologs in a given taxonomic lineage.

Next, the presence/absence matrix of individual ICs was used to perform an orthology profiling clustering and enrichment analysis. Since this analysis is human IC-centric, only the orthologs from eukaryotic lineages were selected. Hierarchical clustering was performed with the Ward method of clustering and Euclidean distance metric using the SciPy package (*Virtanen et al., 2020*) in Python. The resulting dendrogram was used to define nine clusters that group ICs with similar presence/absence patterns. Human ICs falling in these nine clusters were then subjected to a functional enrichment analysis using the Gene Ontology resource GO enrichment tool (*Carbon et al., 2021*). GO terms with a corrected FDR <0.01 were retained as significantly enriched terms.

## CALHM evolutionary analysis

BPPS was used to perform a pattern-based classification of the CALHM homologs. First, orthologs for all six human CALHMs were collected. This ortholog dataset was supplemented with more hits from the UniProt database using MAPGAPS, a multiply-aligned profile for global alignment of protein sequences (*Neuwald, 2009*). Along with finding the best hits for the human CALHMs, MAPGAPS also aligns those hits to the template profile alignment to generate a large multiple-sequence alignment of the resulting 5805 CALHM. This large alignment was then subjected to BPPS, which performs a hierarchical classification of the sequence sets based on conserved pattern positions shared by subsets of sequences using a Bayesian statistical procedure (*Neuwald, 2014*). This generates a hierarchical cluster where sequences within each cluster are defined by distinct conserved patterns.

## Cell lines

HEK293T cells are purchased from Sigma-Aldrich (Catalog Number: 96121229). Neuro2A cells are purchased from ATCC (Catalog Number: CCL-131). The cells are authenticated and tested negative for mycoplasma contamination by the vendor.

## CALHM plasmid construction and expression by transient transfection

Full-length human CALHM1 and CALHM6 in the pEGC Bacmam vector were used. The translated product contains the human CALHM1 or CALHM6 protein, a thrombin digestion site (LVPRGS), an enhanced GFP protein, and an 8× His tag. Primers for site-directed mutagenesis were designed using Snapgene and synthesized by Eurofins Genomics. The QuikChange mutagenesis protocol was used to generate all the mutants of the study. Sanger sequencing was performed to identify positive clones.

Adherent HEK293T (ECACC, Catalog Number: 96121229) cells were grown in DMEM media supplemented with 10% fetal bovine serum. Transient transfection was conducted using Lipofectamine 2000 by following the manufacturer's protocol. Specifically, the cells were cultured in 60 mm Petri dishes until 80% confluency. Transfection solution was made by mixing 500 ng of plasmid DNA, 4 µL of Lipofectamine 2000 reagent, and 100 µL Opti-MEM media. After 20 min incubation at room temperature, the DNA-lipid complexes were added to the cell culture and incubated at 37°C. The next day, 10 mM sodium butyrate was added to the cells to boost protein expression. The cell culture was then grown at 30°C for another day before harvesting. The cell pellet was flash-frozen with liquid nitrogen and stored at –80°C. Each CALHM1 and CALHM6 mutant was transfected in triplicate as biological replicates.

## Expression analysis of CALHM1, CALHM6, and their mutants

To analyze total expression levels of wild-type CALHM1, wild-type CALHM6, and their mutants, cells were lysed in TBS buffer (20 mM Tris, pH 8.0, 150 mM NaCl) supplemented with 10% lauryl maltose neopentyl glycol and cholesterol hemisuccinate detergents on ice. Lysates were solubilized at 4°C for 1 hr and clarified by centrifugation at 13,000 rpm for 5 min. The supernatant was mixed with 4× SDS loading buffer containing 5% 2-mercaptoethanol and resolved on a 4–20% gradient SDS-PAGE gel.

In-gel fluorescence imaging of the C-terminal GFP tag was performed immediately after electrophoresis using a ChemiDoc system to visualize CALHM1 and CALHM6 proteins. Following imaging, proteins were transferred to a nitrocellulose membrane using semi-dry transfer buffer (48 mM Tris base, 39 mM glycine, 20% methanol). Membranes were blocked with TBST (20 mM Tris, pH 8.0, 150 mM NaCl, 0.1% Tween-80) containing 4% non-fat milk for 1 hr at room temperature. β-Actin was detected as a loading control by incubating membranes with HRP-conjugated anti-β-actin antibody (Proteintech, Catalog Number: HRP-60008; 1:2000 dilution) for 1 hr at 4°C. After four washes with TBST (15 min each), chemiluminescence signals were developed using Pierce ECL substrate and imaged on a ChemiDoc system. A brightfield image was overlaid to visualize protein marker positions.

Each mutant was analyzed in triplicate from independently transfected cell pellets. GFP signal intensities for CALHM1 and CALHM6 were quantified using ImageJ and normalized to β-actin chemiluminescence signals. Mean values and SEM were calculated for each mutant, and relative expression levels were compared to wild-type proteins using bar graphs generated in Microsoft Excel.

## Surface expression analysis

Surface biotinylation of CALHM1, CALHM6, and their mutants was performed using the Pierce Cell Surface Biotinylation and Isolation Kit (Thermo Fisher Scientific) following the manufacturer's protocol. Cells were cultured and harvested 48 hr post-transfection as described above. Surface-isolated proteins were analyzed by SDS-PAGE, and in-gel fluorescence imaging of the C-terminal GFP tag was performed using a ChemiDoc system.

## Electrophysiology

TsA201 cells expressing plasmids encoding N-terminal GFP-tagged human CALHM1 or CALHM6 were used. After 1 day post-transfection with plasmid DNA (100 ng/mL) and Lipofectamine 2000 (Invitrogen, 11668019), the cells were trypsinized and replated onto poly-L-lysine-coated (Sigma P4707) glass coverslips. After cell attachment, the coverslip was transferred to a recording chamber. Whole-cell patch-clamp recordings were performed at room temperature (21–23°C) or body temperature (36–38°C). Signals were amplified using a Multiclamp 700B amplifier and digitized using a Digidata 1550B A/D converter (Molecular Devices, Sunnyvale, CA, USA). The whole-cell current was measured on the cells with an access resistance of less than 10 MΩ after the whole-cell configuration was obtained. The amplifier circuitry compensated the whole-cell capacitance. The two-step pulse from 120 mV to –80 mV for 50 ms was continuously applied to the cell membrane every 5 s to monitor the

activation of the CALHM current. The step pulse from –100 mV to 140 mV (or –80 mV to 120 mV) for 200 ms with a holding potential of 0 mV was applied to plot the current-voltage relationship. Electrical signals were digitized at 10 kHz and filtered at 2 kHz. Recordings were analyzed using Clampfit 11.3 (Axon Instruments Inc), GraphPad Prism 10 (La Jolla, CA, USA), and OriginPro 2024 (OriginLab, Northampton, MA, USA). The standard bath solution contains (in mM): 150 NaCl, 5 KCl, 1 $MgCl_2$, 2 $CaCl_2$, 12 Mannitol, 10 HEPES, pH = 7.4 with NaOH. For a whole-cell recording, the extracellular solution contains (in mM): 150 NaCl, 10 HEPES, 1 $MgCl_2$, 5 $CaCl_2$. To establish a zero $Ca^{2+}$ condition, 5 mM $CaCl_2$ was replaced with 5 mM EGTA. For the zero $Ca^{2+}$ condition with 100 µM $Gd^{3+}$, 5 mM $CaCl_2$ was omitted entirely without adding additional EGTA. The intracellular solution contains (in mM): 150 NaCl, 10 HEPES, 1 $MgCl_2$, 5 EGTA.

All data are expressed as mean ± SEM. Multiple comparisons were performed by one-way or two-way ANOVA with Bonferroni's post hoc test. n indicates the number of cells. Significance was defined as: $*p<0.05$, $**p<0.01$, $***p<0.001$. The absence of an asterisk indicates nonsignificance.

## Acknowledgements

We thank members of the Kannan Lab for feedback and rotation student William N Lantz for the manual evaluation of RAG LLM-based annotations. Maya Salcedo is acknowledged for onboarding the undergraduates involved in the project. Funding for NK and WL from NIH Common Fund (IDG) U01CA271376 is acknowledged. ZR is supported by 5R00NS128258 from NINDS.

## Additional information

### Funding

| Funder | Grant reference number | Author |
|---|---|---|
| National Institutes of Health | U01CA271376 | Rahil Taujale<br>Nathan Gravel<br>Saber Soleymani<br>Rayna Carter<br>Kennady Boyd<br>Wei Lü<br>Natarajan Kannan |
| National Institute of Neurological Disorders and Stroke | 5R00NS128258 | Zheng Ruan |

The funders had no role in study design, data collection and interpretation, or the decision to submit the work for publication.

### Author contributions

Rahil Taujale, Conceptualization, Resources, Data curation, Formal analysis, Validation, Investigation, Visualization, Methodology, Writing – original draft, Writing – review and editing; Sung Jin Park, Saber Soleymani, Data curation, Formal analysis, Visualization, Methodology, Writing – original draft; Nathan Gravel, Resources, Formal analysis, Validation, Investigation; Rayna Carter, Data curation, Formal analysis, Methodology; Kennady Boyd, Resources, Data curation; Sarah I Keuning, Formal analysis, Validation; Zheng Ruan, Supervision, Validation, Methodology; Wei Lü, Conceptualization, Funding acquisition, Investigation, Methodology, Writing – original draft, Project administration, Writing – review and editing; Natarajan Kannan, Conceptualization, Supervision, Funding acquisition, Validation, Investigation, Visualization, Methodology, Writing – original draft, Project administration, Writing – review and editing

### Author ORCIDs

Rahil Taujale ⓘD https://orcid.org/0000-0003-1292-1619
Sarah I Keuning ⓘD https://orcid.org/0009-0008-9408-5119
Wei Lü ⓘD https://orcid.org/0000-0002-3009-1025
Natarajan Kannan ⓘD https://orcid.org/0000-0002-2833-8375

Reviewer #1 (Public review): https://doi.org/10.7554/eLife.106134.3.sa1
Reviewer #2 (Public review): https://doi.org/10.7554/eLife.106134.3.sa2
Author response https://doi.org/10.7554/eLife.106134.3.sa3

## Additional files

### Supplementary files

Supplementary file 1. Gene Ontology (GO) annotation of IC clusters. (A) Table showing the enriched GO terms for each IC cluster. An * in the ion and gate mechanism columns indicates annotations not verified by the retrieval augmented generation (RAG) system. (B) Table showing the presence/absence of curated human IC channels in other ion channel databases listed in Kyoto Encyclopedia of Genes and Genomes (KEGG), Guide to Pharmacology (GtoP), and Pharos. (C) List of human IC orthologs detected across the tree of life. (D) Table showing the enriched Gene Ontology (GO) terms for each IC cluster.

MDAR checklist

### Data availability

The human IC annotation table with all the curation information is made available with the manuscript as *Supplementary file 1A*. The fasta sequences for the human ICs (both full-length sequences and the pore domain sequences) are available through Zenodo. The full length sequences for all the identified orthologs are also available through Zenodo. The code and results related to the RAG annotation pipeline are available at GitHub (copy archived at *Kannan, 2026*).

The following dataset was generated:

| Author(s) | Year | Dataset title | Dataset URL | Database and Identifier |
|---|---|---|---|---|
| Taujale R, Kannan N | 2025 | Identification and classification of ion-channels across the tree of life: Insights into understudied CALHM channels | https://doi.org/10.5281/zenodo.16232528 | Zenodo, 10.5281/zenodo.16232528 |

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
