## [Editor Report · eLife Assessment]

In this manuscript Taujale et al describe an interdisciplinary approach to mine the human channelome and further discover orthologues across diverse organisms. Further, this work provides evidence that supports a role for conserved residues in CALHM channel gating. Overall this **important** work presents findings that can be helpful to the ion channel community, as well as to those interested in improved methods for mining sequence space for their protein of interest. However, further validation of the improvements their approach shows over previous approaches is needed, making this a **solid** contribution to the literature in this field.

---

## [Referee Report · Reviewer #1 (Public review)]

Summary:

In the manuscript "Identification and classification of ion-channels across the tree of life: Insights into understudied CALHM channels" Taujale et al describe an interdisciplinary approach to mine the human channelome and further discover orthologues across diverse organisms, culminating in delineating co-conserved patterns in an example ion channel: CALHM. Overall, this paper comes in two sections, one where 419 human ion channels and 48,000+ channels from diverse organisms are found through a multidisciplinary data mining approach, and a second where this data is used to find co-conserved sequences, whose functional significance is validated via experiments on CALHM1 and CALHM6. Overall, this is an intriguing data-first approach to better understand even understudied ion channels like CALHM6. However, more needs to be done to pull this story together into a single coherent narrative.

Strengths:

This manuscript takes advantage of modern-day LLM tools to better mine the literature for ion channel sequences in humans and other species with orthologous ion channel sequences. They explore the 'dark channome' of understudied ion channels to better reveal the information evolution has to tell us about our own proteins, and illustrate the information this provides access to in experimental studies in the final section of the paper. Finally, they provide a wealth of information in the supplementary tables (in the form of Excel spreadsheets and a dataset on Zenodo) for others to explore. Overall, this is a creative approach to a wide-reaching problem that can be applied to other families of proteins.

Weaknesses:

Overall, while a considerable amount of work has been done for this manuscript, the presentation, both in terms of writing and figures, still can use more work even after a first round of revisions. While they have improved their discussion to more clearly describe the need for a better-curated sequence database of ion channels, and how existing resources fall short, some aspects of this process and the motivation remain unclear, especially when it comes to the CALHM sequences.

Overall, this manuscript is a valuable contribution to the field, but requires a few main things to make it truly useful. Namely, how has this approach really improved their ability to identify conserved residues in CALHM over a less-involved approach? And better organization of the first results section of the paper, which is critical to the downstream understanding of the paper, as well as some cosmetic improvements.

---

## [Referee Report · Reviewer #2 (Public review)]

Summary:

In this paper, the authors defined the "channelome," consisting of 419 predicted human ion channels as well as 48,000 ion channel orthologs from other organisms. Using this information, the ion channels were clustered into groups, which can potentially be used to make predictions about understudied ion channels in the groups. The authors then focused on the CALHM ion channel family, mutating conserved residues and assessing channel function.

Strengths:

The curation of the channelome provides an excellent resource for researchers studying ion channels. Supplemental Table 1 is well organized with an abundance of useful information.

Comments on revisions:

The authors have thoroughly addressed my concerns and the manuscript is substantially improved. I have just a few suggestions regarding wording/clarification.

In Supplemental Figure 4, the Western blots (n=3) were quantitated, but the surface biotinylation was not. While I suppose that it is fine to just show one representative experiment for the biotinylation assay, the authors should indicate in the legend how many times this was done. It is essential to know whether these data in Supplemental Figure 4E, F are reproducible as they are absolutely critical for interpretation of all of the data in Figure 5.

---

## [Author Response]

The following is the authors’ response to the original reviews.

**Reviewing Editor Comments:**
(A) Revisions related to the first part, regarding data mining and curation:(1) One question that arises with the part of the manuscript that discusses the identification and classification of ion channels is whether these will be made available to the wider public. For the 419 human sequences, making a small database to share this result so that these sequences can be easily searched and downloaded would be desirable. There are a variety of acceptable formats for this: GitHub/figshare/zenodo/university website that allows a wider community to access their hard work. Providing such a resource would greatly expand the impact of this paper. The same question can be asked of the 48,000+ ion channels from diverse organisms.

We thank the reviewer for providing this important feedback. While the long term plan is to provide access to these sequences and annotations through a knowledge base resource like Pharos, we agree with the comments that it would be beneficial to have these sequences made available with the manuscript as well. We have compiled 3 fasta files containing the following: (1) Full length sequences for the curated 419 ion channel sequences. (2) Pore containing domain sequences for the 343 pore domain containing human ion channel sequences. (3) All the identified orthologs for the human ion channels.

For each sequence in these files, we have extended the ID line to include the most pertinent annotation information to make it readily available. For example, the id>sp|P48995|TRPC1_HUMAN|TRP:VGIC--TRP-TRPC|pore-forming|dom:387-637 provides the classification, unit and domain bounds for the human TRPC1 in the fasta file itself.

These files have been uploaded to Zenodo and are available for download with doi 10.5281/zenodo.16232527. We have included this in the Data Availability statement of the manuscript as well.

(2) Regarding the 48,000+ sequences, what checks have been done to confirm that they all represent bona fide, full-length ion channel sequences? Uniprot contains a good deal of unreviewed sequences, especially from single-celled organisms. The process by which true orthologues were identified and extraneous hits discarded should be discussed in more detail, and all inclusion criteria should be described and justified, clearly illustrating that the risk of gene duplicates and fragments in this final set of ion channel orthologues has been avoided. Related to this, does this analysis include or exclude isoforms?

We thank the reviewer for raising this important point. Our selection of curated proteomes and the KinOrtho pipeline for orthology detection returns, up to an extent, reliable orthologous sequence sets. In brief, our database sequences are retrieved from full proteomes that only include proteins that are part of an official proteome release. Thus, they are mapped from a reference genome to ensure species-specific relevance and avoid redundancy. The >1500 proteomes in this analysis were selected based on their wider use in other orthology detection pipelines like OMA and InParanoid. Our orthology detection pipeline, KinOrtho, performs a fulllength and a domain-based orthology detection which ensures that the orthologous relationships are being defined based on the pore-domain sequence similarity.

But we agree with the reviewer that this might leave room for extraneous, fragments or misannotated sequences to be included in our results. Taking this into careful consideration, we have expanded our sequence validation pipeline to include additional checks such as checking the uniport entry type, protein existence evidence and sequence level checks such as evaluating the compositional bias, non-standard codons and sequence lengths. These validation steps are now described in detail in the Methods section under orthology analysis (lines 768-808). All the originally listed orthologous sequences passed this validation pipeline and thus provide additional confidence that they are bona fide full length ion channel sequences.

We have also expanded this section (lines 758 – 766) to provide more details of the KinOrtho pipeline for orthology detection, which is a previously published method used for orthology detection in kinases by our lab.

Finally, our orthology analysis excludes isoforms and only spans the primary canonical sequences that are part of the UniProt Proteomes annotated sequence set. The isoforms that are generally available in UniProt Proteomes in a separate file named *_additional.fasta were not included in this analysis.

(3) The decision to show the families of ion channels in Figure 1 as pie charts within a UMAP embedding is intriguing but somewhat non-intuitive and difficult to understand. Illustrating these results with a standard tree-like visualization of the relationship of these channels to each other would be preferred.

We appreciate the feedback provided by the reviewer, and understand that a standard tree-like visualization would be much easier to interpret and familiar than a bubble chart based on UMAP embeddings. However, we opted to use the bubble chart for the following reasons:

Low sequence similarity: the 419 human ICs share very minimal sequence similarity, falling in the twilight zone or lower (Dolittle, 1992; PMID:1339026). Thus, traditional multiple sequence alignment and phylogenetic reconstruction methods perform very poorly and generate unreliable or even misleading results. To explore the practicality of this option, we pursued performing a multiple sequence alignment of just 3 of the possibly related IC families as suggested by reviewer 2 (CALHM, Pannexins, and Connexins) using the state of the art structure based sequence alignment method Foldmason (doi: https://doi.org/10.1101/2024.08.01.606130). Even then, the sequence alignment and the resulting tree for just these 3 families were poor and unreliable, as illustrated in the attached Author response Image 2.

Protein embeddings based clustering: Novel LLM based approaches such as the protein language model embeddings offer ways to overcome these limitations by capturing sequence, structure, function and evolutionary properties in a high-dimensional space. Thus, we employed this model using DEDAL followed by UMAP for dimensionality reduction, which preserves biologically meaningful local and global relationships.

Abstraction at family level: In Figure 1, we aggregate individual channels into family bubbles with their positions representing the average UMAP coordinates of their members. This offers a balance between an intuitive view of how IC families are distributed in the embedding space and reflects potential functional and evolutionary proximities, while not being impeded by individual IC relationships across families.

We have revised the figure legend (lines 1221 – 1234) with additional description of the visualization and the process used to generate it, and the manuscript text (lines 248-270) provides the rationale behind the selection of this method.

(4) A strength of this paper is the visualization of 'dark' ion channels. However, throughout the paper, this could be emphasized more as the key advantage of this approach and how this or similar approaches could be used for other families of proteins. Specifically, in the initial statement describing 'light' vs 'dark channels', the importance of this distinction and the historical preference in science to study that which has already been studied can be discussed more, even including references to other studies that take this kind of approach. An example of a relevant reference here is to the Structural Genomics Consortium and its goals to achieve structures of proteins for which functions may not be well-characterized. Clarifying these motivations throughout the entire paper would strengthen it considerably.

We thank the reviewer for this constructive comment and agree that highlighting the strength of visualizing “dark” channels and prioritizing them for future studies would strengthen the paper. As suggested, we have revised the text throughout the paper (lines 84-89, 176-180) to contextualize and emphasize this distinction. We have also added a reference for the Structural Genomics Consortium, which, along with resources like IDG, has provided significant resources for prioritizing understudied proteins.

(5) Since the authors have generated the UMAP visualization of the channome, it would be interesting to understand how the human vs orthologue gene sets compare in this space.

We appreciate the reviewer’s input. It is an interesting idea to explore the UMAP embedding space for the human ICs along with their orthologs. The large number of orthologous sequences (>37,000) would certainly impose a computational challenge to generate embeddings-based pairwise alignments across all of them. Downstream dimensionality reduction from such a large set and the subsequent visualization would also suffer from accuracy and interpretability concerns. However, to follow up on the reviewer’s comments, we selected orthologous sequences from a subset of 12 model organisms spanning all taxa (such as mouse, zebrafish, fruit fly, *C. elegans*, *A. thaliana*, *S. cerevisiae*, *E. coli*, etc.).This increased the number of sequences for analysis to 1094 from 343, which is still manageable for UMAP. Using the exact same method, we generated the UMAP embeddings plot for this set as shown below.

**Author response image 1. sa3fig1:** UMAP embeddings of the human ICs alongside orthologs from 12 model organisms.

As shown above, we observed that each orthologous set forms tight, well-defined clusters, preserving local relationships among closely related sequences. For example, a large number of VGICs cluster more closely together compared to Supplementary Figure 1 (with only the human ICs). However, families that were previously distant from others now appear to be even more scattered or pushed further away, indicating a loss of global structure. This pattern suggests that while local distances are well preserved, the global topology of the embedding space could be compromised. Moreover, we find that the placement of ICs with respect to other families is highly sensitive to the parameter choices (e.g., n_neighbors and min_dist), an issue which we did not encounter when using only the human IC sequences. The inclusion of a large number of orthologous sequences that are highly similar to a single human IC but dissimilar to others skews the embedding space, emphasizing local structure at the expense of global relationships.

Since UMAP and similar dimensionality reduction methods prioritize local over global structure, the resulting embeddings accurately reflect strong ortholog clustering but obscure broader interfamily relationships. Consequently, interpreting the spatial arrangement of human IC families with respect to one another becomes unreliable. We have made this plot available as part of this response, and anyone interested can access this in the response document.

(6) Figure 1 should say more clearly that this is an analysis of the human gene set and include more of the information in the text: 419 human ion channel sequences, 75 sequences previously unidentified, 4 major groups and 55 families, 62 outliers, etc. Clearer visualizations of these categories and numbers within the UMAP (and newly included tree) visualization would help guide the reader to better understand these results. Specifically, which are the 75 previously unidentified sequences?

We thank the reviewer for the comments. To address this, we have revised Figure 1 and added more information, including a clear header that states that these are only human IC sets, numbers showing the total number of ICs, and the number of ICs in each group. We have further included new Supplementary Figure 2 and Supplementary Table 2, which show the overlap of IC sequences across the different resources. Supplementary Figure 2 is an upset plot that provides a snapshot of the overlap between curated human ICs in this study compared to KEGG, GtoP, and Pharos. Supplementary Table 2 provides more details on this overlap by listing, for each human IC, whether they are curated as an IC in the 3 IC annotation resources. We believe these additions should provide all the information, including the unidentified sequences we are adding to this resource.

(7) Overall, the manuscript needs to provide a clearer description of the need for a better-curated sequence database of ion channels, as well as how existing resources fall short.

We thank the reviewer for pointing out this important gap in the description. As suggested, we have revised the text thoroughly in the Introduction section to address this comment. Specifically, we have added sections to describe existing resources at sequence and structure levels that currently provide details and/or classification of human ion channels. Then, we highlight the facts that these resources are missing some characterized pore-containing ICs, do not include any information on auxiliary channels, and lack a holistic evolutionary perspective, which raises the need for a better-curated database of ion channels. Please refer to lines 57-63, 73-79, and 95 – 119 for these changes and additions.

(8) Some of the analysis pipeline is unclear. Specifically, the RAG analysis seems critical, but it is unclear how this works - is it on top of the GPT framework and recursively inquires about the answer to prompts? Some example prompts would be useful to understand this.

We thank the reviewer for highlighting this gap in explanation. We understand that the details provided in the Methods and Supplementary Figure 1 may not have sufficiently explained the pipeline, and are missing some important details. The RAG pipeline leverages vector-based retrieval integrated with OpenAI’s GPT-4o model to systematically search literature and generate evidence-based answers. The process is as follows:

Literature sources (PubMed articles) relevant to the annotated ion channels were converted into vector representations stored in a Qdrant database.

Queries constructed from the annotated IC dataset were submitted to the vector database, retrieving contextually relevant literature segments.

Retrieved contexts served as inputs to the GPT-4o model, which produced structured JSON-formatted responses containing direct evidence regarding ion selectivity and gating mechanisms, along with associated confidence scores.

To clarify this further, we have rewritten the relevant subsection in lines 649 - 718. Now, this section provides a detailed description of the RAG pipeline. Also, we have improved Supplementary Figure 1 to provide a clearer description of the pipeline. We have also provided an example prompt template to illustrate the query. These additions clarify how the pipeline functions and demonstrate its practical utility for IC annotation.

(9) The existence of 76 auxiliary non-pore containing 'ion channel' genes in this analysis is a little confusing, as it seems a part of the pipeline is looking for pore-lining residues. Furthermore, how many of these are picked up in the larger orthologues search? Are these harder to perform checks on to ensure that they are indeed ion channel genes? A further discussion of the choice to include these auxiliary sequences would be relevant. This could just be further discussion of the literature that has decided to do this in the past.

We thank the reviewer for this comment, and agree that further clarification of our selection and definition of auxiliary IC sequences would be helpful. As the reviewer has pointed out, one of the annotation pipeline steps is indeed looking for the pore-lining residues. Any sequences that do not have a pore-containing domain are then considered to be auxiliary, and we search for additional evidence of their binding with one of the annotated pore-containing ICs. If such evidence is not found in the literature, we remove them from our curated IC list.

In response to the above comment, we have revised the manuscript text to provide these details. In the Introduction section, we have added references to previous literature that have described auxiliary ICs and also pointed out that the existing ion channel resources do not account for such auxiliary channels (lines 73-79, 107-108,148-149). We have also expanded the Methods section to describe the selection and definition of auxiliary channels (lines 640-646).

With regards to the orthology analysis, since auxiliary channels do not have a pore domain, and our orthology pipeline requires a pore domain similarity search and hit, we did not include them in this part of the analysis. We have clarified the text in the Results section to ensure this is communicated properly throughout the manuscript (lines 212-215, 260-263).

(10) Why are only evolutionary relationships between rat, mouse, and human shown in Figure 3A? These species are all close on the evolutionary timeline.

We thank the reviewer for this comment. Figure 3A currently provides a high-level evolutionary relationship across the 6 human CALHM members as a pretext for the pattern based Bayesian analysis. However, since this analysis is based on a wider set of orthologs that span taxa, we agree that a larger tree that includes more orthologs is warranted.

We have now revised Figure 3A to include an expanded tree that includes 83 orthologs from all 6 human CALHM members spanning 14 organisms from different taxa, ranging from mammals, fishes, birds, nematodes, and cnidarians. The overall structure of the tree is still consistent with 2 major clades as before, with CALHM 1 and 3 in the first clade and CALHM 2,4,5, and 6 in the second clade, with good branch support.

(B) Revisions related to the second part, regarding the analysis of CAHLM channel mutations:(1) It would strengthen the manuscript if it included additional discussion and references to show that previous methods to analyze conserved residues in CALHM were significantly lacking. What results would previous methods give, and why was this not enough? Were there just not enough identified CALHM orthologues to give strong signals in conservation analysis? Also, the amino acid conservation between CLHM-1 and CALHM1 is extremely low. Thus, there are other CALHM orthologs that give strong signals in conservation analysis. There are ~6 papers that perform in-depth analysis of the role of conserved residues in the gating of CALHM channels (human and *C. elegans*) that were not cited (Ma et al, Am J Physiol Cell Physiol, 2025; Syrjanen et al, Nat Commun, 2023; Danielli et al, EMBO J, 2023; Kwon et al, Mol Cells, 2021; Tanis et al, Am J Physiol Cell Physiol, 2017; Tanis et al, J Neurosci, 2013; Ma et al, PNAS, 2013) - these data needs to be discussed in the context of the present work.

We thank the reviewer for the comment and agree that these are excellent studies that have advanced understanding of conserved residues in CALHM gating. While their analyses compared a limited set of sequences, focusing on residues conserved in specific CALHM homologs or species like *C. elegans*, our analysis encompasses thousands of sequences across the entire CALHM family, allowing us to identify residues conserved across all family members over evolution. We also coupled this sequence analysis with hypotheses derived from our published structural studies (Choi et al., Nature, 2019), which highlighted the NTH/S1 region as a critical element in channel gating. Based on this, we focused on evolutionarily conserved residues in the S1–S2 linker and at the interface of S1 with the rest of the TMD, reasoning that if S1 movement is essential for gating, these two structural elements (acting as a hinge and stabilizing interface, respectively) would be key determinants of the conformational dynamics of S1. These regions have been largely overlooked in previous studies. As a result, the residues highlighted in our study do not overlap with those previously reported but instead provide complementary insights into gating mechanisms in this unique channel family. Together, our study and the published literature suggest that many regions and residues in CALHM proteins are critical for gating: while some are conserved across the entire family evolutionarily, others appear conserved only within certain species or subfamilies.

To address the reviewer’s comment, and to highlight the points mentioned above, we have added a brief discussion of these studies and the relevant citations in the revised manuscript (lines 378– 385, 563–576).

(2) Whereas the current-voltage relations for WT channels are clearly displayed, the data that is shown for the mutants does not allow for determining if their gating properties are indeed different than WT.First, the current amplitudes for the mutants were quantified at just one voltage, which makes it impossible to determine if their voltage-dependence was different than WT, which would be a strong indicator for an effect in gating. Current-voltage relations as done for the WT channels should be included for at least some key mutations, which should include additional relevant controls like the use of Gd3+ as an inhibitor to rule out the contribution of some endogenous currents.

We thank the reviewer for this comment. To address this, we performed additional experiments using a multi-step pulse protocol to obtain current-voltage relations for WT CALHM1, CALHM1(I109W), WT CALHM6, and CALHM6(W113A). Our initial two-step protocol (−80 mV and +120 mV) covers both the physiological voltage range and the extended range commonly used in biophysical characterization of ion channels. Most mutants did not exhibit channel activation even within this broad range. We therefore focused on the three mutants that did show substantial activation to perform full I–V analysis as suggested. In all groups, currents activated at 37 °C were significantly inhibited by Gd^3+^, consistent with published reports (Ma et al., AJP 2025; Danielli et al., EMBO J 2023; Syrjänen et al., Nat Commun 2023). Notably, for CALHM6(Y51A), while this mutation did not significantly alter current amplitudes at positive membrane potentials, it markedly reduced currents at negative potentials, rendering the channel outwardly rectifying and altering its voltage dependence. These new data are incorporated into Figure 5 (panels A–O) and discussed in the manuscript. Figure 5 now also shows current amplitudes at both +120 mV and −80 mV in 0 mM Ca^2+^ at 37 °C to facilitate direct comparison between WT and mutants. The previous data at 5 mM Ca^2+^ and 0 mM Ca^2+^ at 22 °C have been moved to Supplementary Figure 5 as requested.

Second, it is unclear whether the three experimental conditions (5 mM Ca^2+^, and 0 Ca^2+^, at 22 and 37C) were measured in the same cell in each experiment, or if they represent different experiments. This should be clarified. If measurements at each condition were done in the same experiment, direct comparison between the three conditions within each individual experiment could further help identify mutations with altered gating.

We thank the reviewer for pointing this out and apologize for the confusion. All three conditions (5 mM Ca^2+^ at 22 °C, 0 mM Ca^2+^ at 22 °C, and 0 mM Ca^2+^ at 37 °C) were sequentially measured in the same cell within each experiment. The currents were then averaged across cells and plotted for each group.

Third, in line 334, the authors state that "expression levels of wild-type proteins and mutants are comparable." However, Western blots showing CALHM protein abundance (Supplementary Fig. 3) are not of acceptable quality; in the top blot, WT CALHM1 appears too dim, representative blots were not shown for all mutants, and individual data points should be included on the group data quantitation of the blots, together with a statistical test comparing mutants with the WT control.

We thank the reviewer for the comment and agree that representative blots were not shown for all mutants. Supplementary Figure 4 (previously Supplementary Figure 3) has been updated to include representative blots for all mutants, individual data points in the quantification, and statistical tests comparing each mutant to the WT control.

A more serious concern is that the total protein quantitation is not very informative about the functional impact of mutations in ion channels, because mutations can severely impact channel localization in the plasma membrane without reducing the total protein that is translated. In mammalian cells, CALHM6 is localized to intracellular compartments and only translocates to the plasma membrane in response to an activating stimulus (Danielli et al, EMBO J, 2023). Thus, if CALHM6 is only intracellular, the protein amount would not change, but the measured current would. Abundant intracellular CALHM1 has also been observed in mammalian cells transfected with this protein (Dreses-Werringloer et al., Cell, 2008). Quantitation of surface-biotinylated channels would provide information on whether there are differences between the constructs in relation to surface expression rather than gating. An alternative approach to biotinylation would be to express GFP-tagged constructs in Xenopus oocytes and look for surface expression. This is what has been done in previous CALHM channel studies.Without evidence for the absence of defects in localization or clear alterations in gating properties, it is not possible to conclude whether mutant channels have altered activity. Does the analysis of sequences provide any testable hypotheses about substitutions with different side chains at the same position in the sequence?

We thank the reviewer for this very important comment. We agree that total protein levels alone do not distinguish between intracellular retention and proper trafficking to the plasma membrane. To address this, we performed surface biotinylation assays for all WT and mutant CALHM1 and CALHM6 constructs to assess their plasma membrane localization. The results show that mutants have either comparable or substantially higher surface expression levels than WT, consistent with the Western blot data. Together, these findings support our original interpretation that the observed differences in electrophysiological currents are not due to trafficking defects but reflect functional effects. These new data are presented in Supplementary Figure 5.

(3) Line 303 - 13 aligned amino acids were conserved across all CALHM homologs - are these also aligned in related connexin and pannexin families? It is likely that cysteines and proline in TM2 are since CALHM channels overall share a lot of similarities with connexins and pannexins (Siebert et al, JBC, 2013). As in line 207, it would be expected that pannexins, connexins, and CALHM channel families would group together. Related to this, see Line 406 - in connexins, there is also a proline kink in TM2 that may play a role in mediating conformational changes between channel states (Ri et al, Biophysical Journal, 1999). This should be discussed.

We thank the reviewer for the suggestion. We attempted a structure based sequence alignment of representative structures from all 3 families (CALHM, connexins and pannexins), but the resulting alignments are very poor and have a lot of gapped regions, making it very difficult to comment on the similarities mentioned in this comment. This is actually expected, as although CALHM, connexins, and pannexins are all considered “large-pore” channels, the TMD arrangement and conformation of CALHM are distinct from those of connexins and pannexins. Below, we have included a snapshot of the alignment at the conserved cysteine regions of the CALHM homologs, along with the resulting tree, which has very low support values and has difficulty placing the connexins properly, making it difficult to interpret.

**Author response image 2. sa3fig2:** Structure based sequence alignment and phylogenetic analysis of available crystal structures of members from the CALHM, Pannexin and Connexin families. Top: The resulting sequence alignment is very sparse and does not show conservation of residues in the TM regions. The CPC motif with conserved cysteines in CALHM family is shown. Bottom: Phylogenetic tree based on the alignment has low support values making it difficult to interpret.

(4) Line 36 - This work does not have experimental evidence to show that the selected evolutionarily conserved residues alter gating functions.

Our electrophysiology data demonstrate that the selected evolutionarily conserved residues have a major impact on CALHM1 and CALHM6 gating. As shown in Figure 5, mutations at these residues produce two distinct phenotypes: (1) nonconductive channels, and (2) altered voltage dependence, resulting in outward rectification. Importantly, these functional changes occur despite normal total expression and surface trafficking, as confirmed by Western blotting and surface biotinylation (Supplementary Figure 4). These findings indicate that the affected residues are critical for the conformational dynamics underlying channel gating rather than for protein expression or localization.

(5) Line 296-297 - This could also be put in the context of what we already know about CALHM gating. While all cryo EM structures of CALHM channels are in the open state, we still do understand some things about gating mechanism (Tanis et al Am J Physiol Cell Physiol, Cell Physiol 2017; Ma et al Am J Physiol Cell Physiol, Cell Physiol 2025) with the NT modulating voltage dependence and stabilizing closed channel states and the voltage dependent gate being formed by proximal regions of TM1.

Thank you for providing this suggestion. As suggested, we have revised the text to place our findings in the context of current knowledge about CALHM gating and have added the relevant citations (lines 370-373).

(6) Lines 314-315 - Just because residues are conserved does not mean that they play a role in channel gating. These residues could also be important for structure, ion selectivity, etc.

We agree that evolutionary conservation alone does not imply a role in gating. However, our hypothesis derives from the positioning of these conserved residues, and previous studies that have indicated the importance of the NTH/S1 region for channel gating function. More importantly, our electrophysiology data indicate that these conserved residues specifically impact channel gating in CALHM1 and CALHM6. We have revised the text in lines 404-406 to clarify this further.

(7) Line 333 - while CALHM6 is less studied than CALHM1, there is knowledge of its function and gating properties. Should CALHM6 be considered a "dark" channel? The IDG development level in Pharos is Tbio. There have been multiple papers published on this channel (ex: Ebihara et al, J Exp Med, 2010; Kasamatsu et al, J Immunol 2014; Danielli et al, EMBO J, 2023).

We thank the reviewer for noting this important discrepancy. We have updated the text and labels related to CALHM6 to reflect its status as Tbio in the manuscript.

(8) Please cite Jeon et al., (Biochem Biophys Res Commun, 2021), who have already shown temperature-dependence of CALHM1.

Thank you for the comment. We have added the citation.

(9) It would be helpful to have a schematic showing amino acid residues, TM domains, highlighted residues mutated, etc.

Thank you for the suggestion. We have revised the figure and added labels for the TM domains, and highlighted the mutated residues.

**Reviewer #1 (Recommendations for the authors):**
(1) Why in the title is 'ion-channels' hyphenated but in the text it is not?

This has been changed.

(2) Line 78: 'Cryo-EM' is not defined before the acronym is used.

This has been fixed.

(3) Typo in line 519: KinOrthto.

This has been fixed.

(4) Capitalizing 'Tree of Life' is a bit strange in section 2 of the results and the Discussion.

We have removed the capitalization as suggested.

(5) In Figure 3 and Supplementary Figure 4A, the gene names in the tree are CAHM and not CALHM - I assume this is an error.

This has been made consistent to CALHM.

(6) Font sizes throughout all figures, with the exception of Figure 1, need to be more legible. The X-axis labels in Figure 2A are hard to read, for example (though I can see that there is also the CAHM/CALHM typo here...). A good rule of thumb is that they should be the same size as the manuscript text. Furthermore, the grey backgrounds of Figure 4 and Figure 5 are off-putting; just having a white background here should be sufficient.

This has been addressed. We have increased the font size in all figures with these revisions. The styling for Figure 4 and 5 has also been made consistent with other figures.

**Reviewer #2 (Recommendations for the authors):**
(1) Line 36 - This work does not have experimental evidence to show that the selected evolutionarily conserved residues alter gating functions.

Addressed in comment #4 for Part B Revisions related to the second part, regarding the analysis of CAHLM channel mutations above.

(2) Line 168 - should also be Supplemental Table 1.

This has been addressed.

(3) Line 170 - 419 human ion channel sequences were identified and this was an increase of 75 sequences over previous number. Which 75 proteins are these?

This is now shown in Supplementary Figure 2 and Supplementary Table 2. Supplementary Figure 2 shows an Upset plot with the number of sequences that overlap across databases and the novel sequences that we have added as part of this study. The 75 specifically refers to the sequences that were not included in Pharos, which was chosen to refer to this number since it has the highest number of ICs listed out of all the other resources. Further, Supplementary Table 2 now provides a list of individual ICs and whether they were present in each of the 3 databases compared.

(4) Line 289 - Ca2+ (not Ca); other similar mistakes throughout the manuscript

These have been fixed.

(5) Line 291-292 - Please include more about functions for CALHM channels; ex. CALHM1 regulates cortical neuron excitability (Ma et al, PNAS 2012), CLHM-1 regulates locomotion and induces neurodegeneration in *C. elegans* (Tanis et al. Journal of Neuroscience 2013); see above for references on CALHM6 function.

We have added the functions as suggested.

(6) Line 296-297 - This could also be put in the context of what we already know about CALHM gating. While all cryo EM structures of CALHM channels are in the open state, we still do understand some things about gating mechanism (Tanis et al Am J Physiol Cell Physiol, Cell Physiol 2017; Ma et al Am J Physiol Cell Physiol, Cell Physiol 2025) with the NT modulating voltage dependence and stabilizing closed channel states and the voltage dependent gate being formed by proximal regions of TM1.

Addressed in comment #5 for Part B Revisions related to the second part, regarding the analysis of CAHLM channel mutations above.

(7) Lines 314-315 - Just because residues are conserved does not mean that they play a role in channel gating. These residues could also be important for structure, ion selectivity, etc.

Addressed in comment #6 for Part B Revisions related to the second part, regarding the analysis of CAHLM channel mutations above.

(8) Line 333 - While CALHM6 is less studied than CALHM1, there is knowledge of its function and gating properties. Should CALHM6 be considered a "dark" channel? The IDG development level in Pharos is Tbio. There have been multiple papers published on this channel (ex: Ebihara et al, J Exp Med, 2010; Kasamatsu et al, J Immunol 2014; Danielli et al, EMBO J, 2023).

Addressed in comment #7 for Part B Revisions related to the second part, regarding the analysis of CAHLM channel mutations above.

(9) Line 627 - Do you mean that 5 mM CaCl2 was replaced with 5 mM EGTA in 0 Ca2+ solution?

This is correct.

(10) Why are only evolutionary relationships between rat, mouse, and human shown in Figure 3A? These species are all close on the evolutionary timeline.

Addressed in comment #10 for Part A Revisions related to the first part, regarding data mining and curation above.

(11) Figure 5 - no need to show the currents at room temperature in the main text since there are robust currents at 37 degrees; this could go into the supplement. Also, please cite Jeon et al. (Biochem Biophys Res Commun, 2021), who have already shown temperature-dependence of CALHM1.

Addressed in comment #8 for Part B Revisions related to the second part, regarding the analysis of CAHLM channel mutations above.

(12) It would be helpful to have a schematic showing amino acid residues, TM domains, highlighted residues mutated etc.

Addressed in comment #9 for Part B Revisions related to the second part, regarding the analysis of CAHLM channel mutations above.

(13) Use of S1-S4 to refer to the transmembrane "segments" is not standard; rather, TM1-TM4 would generally be used to refer to transmembrane domains.

We have used the S1–S4 helix notation to maintain consistency with the nomenclature employed in our previous study (Choi et al., Nature, 2019).